# UNSURE: SELF-SUPERVISED LEARNING WITH UNKNOWN NOISE LEVEL AND STEIN'S UNBIASED RISK ESTIMATE

**Julián Tachella**[*]
CNRS & ENS Lyon
Lyon, France
julian.tachella@cnrs.fr

**Mike Davies**
University of Edinburgh
Edinburgh, UK
mike.davies@ed.ac.uk

**Laurent Jacques**
UCLouvain
Louvain-la-Neuve, Belgium
laurent.jacques@uclouvain.be

## ABSTRACT

Recently, many self-supervised learning methods for image reconstruction have been proposed that can learn from noisy data alone, bypassing the need for ground-truth references. Most existing methods cluster around two classes: i) Stein's Unbiased Risk Estimate (SURE) and similar approaches that assume full knowledge of the noise distribution, and ii) Noise2Self and similar cross-validation methods that require very mild knowledge about the noise distribution. The first class of methods tends to be impractical, as the noise level is often unknown in real-world applications, and the second class is often suboptimal compared to supervised learning. In this paper, we provide a theoretical framework that characterizes this expressivity-robustness trade-off and propose a new approach based on SURE, but unlike the standard SURE, does not require knowledge about the noise level. Throughout a series of experiments, we show that the proposed estimator outperforms other existing self-supervised methods on various imaging inverse problems.

## 1 INTRODUCTION

Learning-based reconstruction methods have recently shown state-of-the-art results in a wide variety of imaging inverse problems, from medical imaging to computational photography (Ongie et al., 2020). However, most methods rely on ground-truth reference data for training, which is often expensive or even impossible to obtain (eg. in medical and scientific imaging applications). This limitation can be overcome by employing self-supervised learning losses, which only require access to noisy (and possibly incomplete) measurement data (Tachella et al., 2023a).

Self-supervised denoising methods cluster around two classes: i) SURE (Metzler et al., 2020), Noisier2Noise (Moran et al., 2020) and similar methods which assume full knowledge about the noise distribution, and ii) Noise2Self (Batson & Royer, 2019) and other cross-validation methods which only require independence of the noise across pixels. While networks trained via SURE perform better than cross-validation methods, they are brittle to misspecification of the noise level, which is often not fully known in real-world imaging settings. This phenomenon can be seen as a robustness-expressivity trade-off, where leveraging more information about the noise distribution results in more optimal estimators, which are at the same time less robust to errors about the noise model. In this paper, we show that this trade-off can be understood through the set of constraints imposed on the derivatives of the estimator: SURE-like estimators do not impose any constraints on derivatives, whereas cross-validation methods add strong constraints on them. Our analysis paves the way for a

---

[*]Code associated to this paper is available at github.com/tachella/unsure.

new family of self-supervised estimators, which we name UNSURE, which lie between these two extremes, by only constraining the expected divergence of the estimator to be zero.

The contributions of the paper are the following:

1. We present a theoretical framework for understanding the robustness-expressivity trade-off of different self-supervised learning methods.
2. We propose a new self-supervised objective that extends the SURE loss for the case where the noise level is unknown and provide generalizations to spatially correlated Gaussian noise (with unknown correlation structure), Poisson-Gaussian noise, and certain noise distributions in the exponential family.
3. Throughout a series of experiments in imaging inverse problems, we demonstrate state-of-the-art self-supervised learning performance in settings where the noise level (or its spatial correlation) is unknown.

## 1.1 RELATED WORK

**SURE**    Stein's unbiased risk estimate is a popular strategy for learning from noisy measurement data alone, which requires knowledge of the noise distribution (Stein, 1981; Efron, 2004; Aggarwal et al., 2023). The SURE loss has been extended for large classes of noise distributions (Hudson, 1978; Raphan & Simoncelli, 2011; Le Montagner et al., 2014), and has been used to train networks in a self-supervised way when the noise distribution is known (Metzler et al., 2020; Chen et al., 2022). ENSURE (Aggarwal et al., 2023), despite having a similar name to this work, proposes a weighting correction to SURE in the case where we obtain observations from multiple operators, and does not handle unknown noise levels. Zhussip et al. (2019) presents a variant of SURE that takes into account two noisy images instead of a single one, and assumes that the noise level is known. SURE belongs more broadly to the class of empirical Bayes methods (Raphan & Simoncelli, 2011; Efron, 2011; Robbins, 1964; Efron, 2012), which build estimators from measurement data alone for a large class of noise distributions, however, to the best of our knowledge, the case of partially unknown noise distribution has not been considered.

**Noise2Noise**    It is possible to learn an estimator in a self-supervised way if two independent noisy realizations of the same image are available for training, without explicit knowledge of the noise distribution (Mallows, 1973). This idea was popularized in imaging by the Noise2Noise method (Lehtinen et al., 2018). However, obtaining two independent noise realizations of the same signal is often impossible.

**Noisier2Noise and Related Methods**    Noisier2Noise (Moran et al., 2020), the coupled bootstrap estimator (Oliveira et al., 2022) and Recorrupted2Recorrupted (R2R) Pang et al. (2021), leverage the fact that two independent noisy images can be obtained from a single noisy image by adding additional noise. However, the added noise must have the same noise covariance as the original noise in the image, and thus full information about the noise distribution is required. Noise2Score (Kim & Ye, 2021), propose to learn the score of the noisy data distribution, and then denoise the images via Tweedie's formula which requires knowledge about the noise level. (Kim et al., 2022) proposes a way to estimate the noise level using the learned score function. However, the method relies on a Noisier2Noise approximation, whereas we provide closed-form formulas for the noise level estimation.

**Noise2Void and Cross-Validation Methods**    When the noise is iid, a denoiser that does not take into account the input pixel to estimate the denoised version of the pixel cannot overfit the noise. This idea goes back to cross-validation-based estimators (Efron, 2004). One line of work (Krull et al., 2019) leverages this idea to build self-supervised losses that remove the center pixel from the input, whereas another line of work builds network architectures whose output does not depend on the center input pixel (Laine et al., 2019). These methods require mild knowledge about the noise distribution (Batson & Royer, 2019) (in particular, that the distribution is independent across pixels), but often provide suboptimal performances due to the discarded information. Several extensions have been proposed: Neighbor2Neighbor (Huang et al., 2021) is among the best-performing for denoising, and SSDU (Yaman et al., 2020) and Noise2Inverse (Hendriksen et al., 2020) generalize the idea for linear inverse problems.

**Learning From Incomplete Data** In many inverse problems, such as sparse-angle tomography, image inpainting and magnetic resonance imaging, the forward operator is incomplete as there are fewer measurements than pixels to reconstruct. In this setting, most self-supervised denoising methods fail to provide information in the nullspace of the forward operator. There are two ways to overcome this limitation (Tachella et al., 2023a): using measurements from different forward operators (Bora et al., 2018; Tachella et al., 2022; Yaman et al., 2020; Daras et al., 2024), or leveraging the invariance of typical signal distributions to rotations and/or translations (Chen et al., 2021; 2022).

**Approximate Message Passing** The approximate message-passing framework (Donoho et al., 2009) for compressed sensing inverse problems relies on an Onsager correction term, which results in divergence-free denoisers in the large system limit (Xue et al., 2016; Ma & Ping, 2017; Skuratovs & Davies, 2021). However, different from the current study, the purpose of the Onsager correction is to ensure that the output error at each iteration appears uncorrelated in subsequent iterations. In this work, we show that optimal self-supervised denoisers that are blind to noise level are divergence-free in expectation.

## 2 SURE AND CROSS-VALIDATION

We first consider the Gaussian denoising problems of the form

$$y = x + \sigma\epsilon \tag{1}$$

where $y \in \mathbb{R}^n$ are the observed measurements, $x \in \mathbb{R}^n$ is the image that we want to recover, $\epsilon \sim \mathcal{N}(0, I)$ is the Gaussian noise affecting the measurements. This problem can be solved by learning an estimator from a dataset of supervised data pairs $(x, y)$ and minimizing the following supervised loss

$$\arg\min_f \mathbb{E}_{x,y}\|f(y) - x\|^2 \tag{2}$$

whose minimizer is the minimum mean squared error (MMSE) estimator $f(y) = \mathbb{E}\{x|y\}$. However, in many real-world applications, we do not have access to ground-truth data $x$ for training, and instead only a dataset of noisy measurements $y$. SURE (Stein, 1981) provides a way to bypass the need for ground-truth references since we have that[1]

$$\mathbb{E}_{x,y}\|f(y) - x\|^2 = \mathbb{E}_y\left[\|f(y) - y\|^2 + 2\sigma^2\,\mathrm{div}f(y) - n\sigma^2\right]. \tag{3}$$

where the divergence is defined as $\mathrm{div}f(y) := \sum_{i=1}\frac{\partial f_i}{\partial y_i}(y)$. Thus, we can optimize the following self-supervised loss

$$\arg\min_{f\in\mathcal{L}^1} \mathbb{E}_y\left[\|f(y) - y\|^2 + 2\sigma^2\,\mathrm{div}f(y)\right] \tag{SURE}$$

where $\mathcal{L}^1$ is the space of weakly differentiable functions, and whose solution is given by Tweedie's formula, i.e.,

$$\mathbb{E}\{x|y\} = y + \sigma^2\nabla\log p_y(y) \tag{4}$$

where $p_y$ is the distribution of the noise data $y$. While this approach removes the requirement of ground-truth data, it still requires knowledge about the noise level $\sigma^2$, which is unknown in many applications.

Noise2Void (Krull et al., 2019), Noise2Self (Batson & Royer, 2019) and similar cross-validation (CV) approaches do not require knowledge about the noise level $\sigma$ by using estimators whose $i$th output $f_i(y)$ does not depend on the $i$th input $y_i$, which in turns implies that $\frac{\partial f_i}{\partial y_i}(y) = 0$ for all pixels $i = 1, \ldots, n$ and all $y \in \mathbb{R}^n$. These estimators have thus zero divergence $\mathrm{div}f(y) = 0$ for all $y \in \mathbb{R}^n$, and thus can be learned by simply minimizing a data consistency term

$$f^{\mathrm{CV}} = \arg\min_{f\in\mathcal{S}_{\mathrm{CV}}} \mathbb{E}_y\|f(y) - y\|^2 \tag{5}$$

---

[1]In practice, we replace the expectations over $y$ by a sum over a finite dataset $\{y_i\}_{i=1}^N$, obtaining empirical estimators. The theoretical analysis focuses on the asymptotic case of $N \to \infty$ to provide a stronger characterization of the resulting estimators.

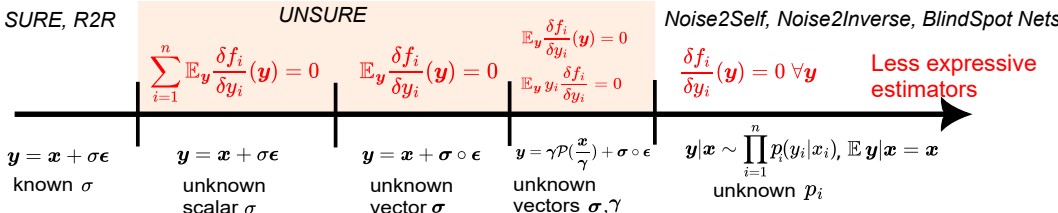

**Figure 1: The expressivity-robustness trade-off in self-supervised denoising.** If the noise distribution is fully known, SURE provides the most expressive estimator, matching the performance of supervised learning. As the assumptions on the noise are relaxed, the learned estimator needs to be less expressive to avoid over-fitting the noise. In this work, we show that popular 'Noise2x' strategies impose too restrictive conditions on the learned denoiser, and propose an alternative that strikes a better trade-off.

where the function is restricted to the space

$$\mathcal{S}_{\text{CV}} = \{f \in \mathcal{L}^1 : \frac{\partial f_i}{\partial y_i}(\boldsymbol{y}) = 0 \text{ for almost every } \boldsymbol{y} \in \mathbb{R}^n\}.$$

Using the SURE identity in (3), we have the equivalence with the following constrained supervised loss

$$f^{\text{CV}} = \arg\min_{f \in \mathcal{S}_{\text{CV}}} \mathbb{E}_{\boldsymbol{x},\boldsymbol{y}} \|f(\boldsymbol{y}) - \boldsymbol{x}\|^2$$

whose solution is $f_i^{\text{CV}}(\boldsymbol{y}) = \mathbb{E}\{y_i|\boldsymbol{y}_{-i}\} = \mathbb{E}\{x_i|\boldsymbol{y}_{-i}\}$ for $i = 1, \ldots, n$ where $\boldsymbol{y}_{-i}$ denotes $\boldsymbol{y}$ excluding the $i$th entry $y_i$. The constraint can be enforced in the architecture using convolutional networks whose receptive field does not look at the center pixel (Laine et al., 2019), or via training, by choosing a random set of pixels at each training iteration, setting random values (or the value of one of their neighbors) at the input of the network and computing the loss only at those pixels (Krull et al., 2019).

The zero derivative constraint results in suboptimal estimators, since the $i$th measurement $y_i$ generally carries significant information about the value of $x_i$. The expected mean squared error of the cross-validation estimator is

$$\mathbb{E}_{\boldsymbol{x},\boldsymbol{y}} \frac{1}{n} \|f^{\text{CV}}(\boldsymbol{y}) - \boldsymbol{x}\|^2 = \frac{1}{n} \sum_{i=1}^{n} \mathbb{E}_{\boldsymbol{y}_{-i}} \mathbb{V}\{x_i|\boldsymbol{y}_{-i}\} \tag{6}$$

with $\mathbb{V}$ the variance operator and the performance of this estimator can be arbitrarily bad if there is little correlation between pixels, ie. $\mathbb{V}\{x_i|\boldsymbol{y}_{-i}\} \approx \mathbb{V}\{x_i\}$, in which case the mean squared error is simply the average variance of the ground-truth data $\mathbb{E}_{\boldsymbol{x},\boldsymbol{y}} \frac{1}{n} \|f^{\text{CV}}(\boldsymbol{y}) - \boldsymbol{x}\|^2 = \frac{1}{n} \sum_{i=1}^{n} \mathbb{V}\{x_i\}$.

## 3 UNSURE

In this work, we propose to relax the zero-derivative constraint of cross-validation, by only requiring that the estimator has *zero expected divergence* (ZED), ie. $\mathbb{E}_{\boldsymbol{y}} \operatorname{div} f(\boldsymbol{y}) = 0$. We can then minimize the following problem:

$$f^{\text{ZED}} = \arg\min_{f \in \mathcal{S}_{\text{ZED}}} \mathbb{E}_{\boldsymbol{y}} \|f(\boldsymbol{y}) - \boldsymbol{y}\|^2 \tag{7}$$

where $\mathcal{S}_{\text{ZED}} = \{f \in \mathcal{L}^1 : \mathbb{E}_{\boldsymbol{y}} \operatorname{div} f(\boldsymbol{y}) = 0\}$. Since we have that $\mathcal{S}_{\text{CV}} \subset \mathcal{S}_{\text{ZED}}$, the estimator that is divergence-free in expectation is more expressive than the cross-validation counterpart. Again due to SURE, (7) is equivalent to the following constrained supervised loss

$$f^{\text{ZED}} = \arg\min_{f \in \mathcal{S}_{\text{ZED}}} \mathbb{E}_{\boldsymbol{x},\boldsymbol{y}} \|f(\boldsymbol{y}) - \boldsymbol{x}\|^2 \tag{8}$$

The constrained self-supervised loss in (7) can be formulated using a Lagrange multiplier $\eta \in \mathbb{R}$ as

$$\min_f \max_\eta \mathbb{E}_{\boldsymbol{y}} \big[ \|f(\boldsymbol{y}) - \boldsymbol{y}\|^2 + 2\eta \operatorname{div} f(\boldsymbol{y}) \big] \tag{UNSURE}$$

which has a simple closed-form solution:

**Proposition 1.** *The solution of* (UNSURE) *is given by*

$$f^{\text{ZED}}(\boldsymbol{y}) = \boldsymbol{y} + \hat{\eta}\nabla \log p_{\boldsymbol{y}}(\boldsymbol{y}) \tag{9}$$

*where* $\hat{\eta} = (\frac{1}{n}\mathbb{E}_{\boldsymbol{y}}\|\nabla \log p_{\boldsymbol{y}}(\boldsymbol{y})\|^2)^{-1} \in \mathbb{R}_+$ *is the optimal multiplier.*

A natural question at this point is, how good can a ZED denoiser be? The theoretical performance of an optimal ZED denoiser is provided by the following result:

**Proposition 2.** *The mean squared error of the optimal divergence-free in expectation denoiser is given by*

$$\mathbb{E}_{\boldsymbol{x},\boldsymbol{y}} \frac{1}{n}\|f^{\text{ZED}}(\boldsymbol{y}) - \boldsymbol{x}\|^2 = \sigma^2(\frac{1}{1 - \frac{\text{MMSE}}{\sigma^2}} - 1) \tag{10}$$

*where* $\text{MMSE} = \mathbb{E}_{\boldsymbol{x},\boldsymbol{y}} \frac{1}{n}\|\mathbb{E}\{\boldsymbol{x}|\boldsymbol{y}\} - \boldsymbol{x}\|^2$ *is the minimum mean squared error.*

The proofs of both propositions are included in Appendix A.

We can derive several important observations from these propositions:

1. As with Tweedie's formula, the optimal divergence-free in expectation estimator in (9) can also be interpreted as doing a gradient descent step on $\log p_{\boldsymbol{y}}$, but the formula is agnostic to the noise level $\sigma$, as one only requires knowledge of $\nabla \log p_{\boldsymbol{y}}(\boldsymbol{y})$ to compute the estimator.

2. The step size $\hat{\eta}$ is a conservative estimate of the noise level, as we have that

$$\hat{\eta} = \frac{\sigma^2}{1 - \frac{\text{MMSE}}{\sigma^2}} \geq \sigma^2 \tag{11}$$

3. As we have that $\mathbb{E}_{\boldsymbol{y}}\|\nabla \log p_{\boldsymbol{y}}(\boldsymbol{y})\|^2 = -\sum_i \mathbb{E}_{\boldsymbol{y}} \frac{\partial^2 \log p_{\boldsymbol{y}}(\boldsymbol{y})}{\partial y_i^2}$, (9) ressembles the natural gradient descent algorithm (Amari, 2016).

4. Combining (4) and (9), we can write $f^{\text{ZED}}$ as the convex combination[2] of the MMSE estimator and the noisy input:

$$f^{\text{ZED}}(\boldsymbol{y}) = \omega\mathbb{E}\{\boldsymbol{x}|\boldsymbol{y}\} + (1 - \omega)\boldsymbol{y} \tag{12}$$

where $\omega = \frac{n\sigma^2}{\mathbb{E}_{\boldsymbol{y}}\|\mathbb{E}\{\boldsymbol{x}|\boldsymbol{y}\} - \boldsymbol{y}\|^2} \in [0, 1]$.

5. We can expand the formula in (10) as a geometric series since $\frac{\text{MMSE}}{\sigma^2} < 1$ to obtain

$$\mathbb{E}_{\boldsymbol{x},\boldsymbol{y}} \frac{1}{n}\|f^{\text{ZED}}(\boldsymbol{y}) - \boldsymbol{x}\|^2 = \text{MMSE} + \sigma^2 \sum_{j=2}^{\infty} (\frac{\text{MMSE}}{\sigma^2})^j. \tag{13}$$

If the support of the signal distribution $p(\boldsymbol{x})$ has dimension $k \ll n$, we have that $\text{MMSE} \approx \sigma^2 k/n$ (Chandrasekaran & Jordan, 2013), and thus i) the estimation of the noise level is accurate as $\hat{\eta} \approx \sigma^2 + (\frac{\sigma k}{n})^2$, and ii) the ZED estimator performs similarly to the minimum mean squared one, $\mathbb{E}_{\boldsymbol{x},\boldsymbol{y}} \frac{1}{n}\|f^{\text{ZED}}(\boldsymbol{y}) - \boldsymbol{x}\|^2 \approx \text{MMSE} + (\frac{\sigma k}{n})^2$. Perhaps surprisingly, in this case, the ZED estimator is close to optimal without adding any explicit constraints about the low-dimensional nature of $p_{\boldsymbol{x}}$.

The ZED estimator fails with high-entropy distributions, for example $p_{\boldsymbol{x}} = \mathcal{N}(\boldsymbol{0}, \boldsymbol{I}\sigma_x^2)$. In this case, we have $\hat{\eta} = \sigma^2 + \sigma_x^2$, as the noise level estimator cannot tell apart the variance coming from the ground truth distribution $\sigma_x^2$ from that coming from the noise $\sigma^2$, and the ZED estimator is the trivial guess $f^{\text{ZED}}(\boldsymbol{y}) = \boldsymbol{0}$. Fortunately, this worst-case setting is not encountered in practice as most natural signal distributions are low-dimensional. To illustrate this, we evaluate the performance of a pretrained deep denoiser in comparison with its ZED version on the DIV2K dataset (see Figure 2). Remarkably, the ZED denoiser performs very similarly to the original denoiser (less than 1 dB difference across most noise levels), and its error is very well approximated by Proposition 2.

Appendix D presents the ZED estimators associated to popular separable signal distributions: spike-and-slab, Gaussian, and two deltas, illustrating the differences with MMSE and cross-validation estimators.

---

[2]The cross-validation estimator can be written as a convex combination of MMSE evaluations, as we have that $f_i^{\text{CV}}(\boldsymbol{y}) = \int_{y_i} \mathbb{E}\{\boldsymbol{x}|\boldsymbol{y}\}p_{y_i}(y_i)dy_i$ for $i = 1, \ldots, n$.

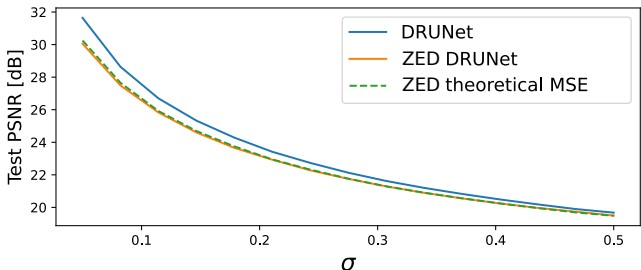

Figure 2: **Removing the expected divergence of pretrained MMSE denoisers.** Considering the DRUNet as an approximate MMSE denoiser, we use the formula in (12) to evaluate its ZED version and plot the expected theoretical error according to (10). The ZED correction only results in a loss of less than 1 dB for most noise levels. Moreover, the theoretical error follows very closely the empirical one.

### 3.1 BEYOND ISOTROPIC GAUSSIAN NOISE

**Correlated Gaussian noise**  In many applications, the noise might be correlated across different pixels. For example, the most popular noise model is

$$\boldsymbol{y}|\boldsymbol{x} \sim \mathcal{N}(\boldsymbol{x}, \boldsymbol{\Sigma})$$

where $\boldsymbol{\Sigma}$ is the covariance matrix capturing the correlations, which is often partially unknown. This setting is particularly challenging where most self-supervised methods fail. If the noise covariance is known, the SURE formula in (3) is generalized as

$$\mathbb{E}_{\boldsymbol{x},\boldsymbol{y}}\|f(\boldsymbol{y}) - \boldsymbol{x}\|^2 = \mathbb{E}_{\boldsymbol{y}}\left[\|f(\boldsymbol{y}) - \boldsymbol{y}\|^2 + 2\operatorname{tr}\left(\boldsymbol{\Sigma}\frac{\partial f}{\partial \boldsymbol{y}}(\boldsymbol{y})\right) - n\operatorname{tr}(\boldsymbol{\Sigma})\right].$$

If the exact form of the noise covariance is unknown, we can consider an $s$-dimensional set of *plausible* covariance matrices

$$\mathcal{R} = \{\boldsymbol{\Sigma}_{\boldsymbol{\eta}} \in \mathbb{R}^{n \times n} : \boldsymbol{\Sigma}_{\boldsymbol{\eta}} = \sum_{j=1}^{s} \eta_j \boldsymbol{\Psi}_j, \boldsymbol{\eta} \in \mathbb{R}^s\}$$

for some basis matrices $\{\boldsymbol{\Psi}_j \in \mathbb{R}^{n \times n}\}_{j=1}^{s}$, with the hope that the true covariance belongs to this set, that is $\boldsymbol{\Sigma} \in \mathcal{R}$. This is equivalent to restricting the learning to estimators in the set $\{f \in \mathcal{L}^1 : \mathbb{E}_{\boldsymbol{y}}\operatorname{tr}\left(\boldsymbol{\Psi}_i\frac{\partial f}{\partial \boldsymbol{y}}(\boldsymbol{y})\right) = 0, \ i = 1, \ldots, s\}$. Here, the dimension $s \geq 1$ offers a trade-off between optimality of the resulting estimator and robustness to a misspecified covariance. We can thus generalize (UNSURE) as

$$\min_{f} \max_{\boldsymbol{\eta}} \mathbb{E}_{\boldsymbol{y}}\|f(\boldsymbol{y}) - \boldsymbol{y}\|^2 + 2\operatorname{tr}\left(\boldsymbol{\Sigma}_{\boldsymbol{\eta}}\frac{\partial f}{\partial \boldsymbol{y}}(\boldsymbol{y})\right). \tag{14}$$

with Lagrange multipliers $\boldsymbol{\eta} \in \mathbb{R}^s$. The following theorem provides the solution for this learning problem:

**Theorem 3.** *Let $p_{\boldsymbol{y}} = p_{\boldsymbol{x}} * \mathcal{N}(\boldsymbol{0}, \boldsymbol{\Sigma})$ and assume that $\{\boldsymbol{\Psi}_j \in \mathbb{R}^{n \times n}\}_{j=1}^{s}$ are linearly independent. The optimal solution of problem* (14) *is given by*

$$f(\boldsymbol{y}) = \boldsymbol{y} + \boldsymbol{\Sigma}_{\hat{\boldsymbol{\eta}}}\nabla \log p_{\boldsymbol{y}}(\boldsymbol{y}) \tag{15}$$

*where $\hat{\boldsymbol{\eta}} = \boldsymbol{Q}^{-1}\boldsymbol{v}$, with $Q_{i,j} = \operatorname{tr}\left(\boldsymbol{\Psi}_i \mathbb{E}_{\boldsymbol{y}}\left[\nabla \log p_{\boldsymbol{y}}(\boldsymbol{y})\nabla \log p_{\boldsymbol{y}}(\boldsymbol{y})^{\top}\right]\boldsymbol{\Psi}_j^{\top}\right)$ and $v_i = \operatorname{tr}(\boldsymbol{\Psi}_i)$ for $i, j = 1, \ldots, s$.*

The proof is included in Appendix A. Proposition 1 is a special case with $s = 1$ and $\boldsymbol{\Psi}_1 = \boldsymbol{I}$.

For example, an interesting family of estimators that are more flexible than $f^{\text{CV}}$ and can handle a different unknown noise level per pixel, is obtained by considering the diagonal parameterization $\boldsymbol{\Sigma}_{\boldsymbol{\eta}} = \operatorname{diag}(\boldsymbol{\eta})$. In this case, we have $\hat{\eta}_i = (\mathbb{E}_{\boldsymbol{y}}\frac{\partial \log p_{\boldsymbol{y}}}{\partial y_i}(\boldsymbol{y})^2)^{-1}$ for $i = 1, \ldots, n$, and thus

$$f_i(\boldsymbol{y}) = y_i + (\mathbb{E}_{\boldsymbol{y}}\frac{\partial \log p_{\boldsymbol{y}}}{\partial y_i}(\boldsymbol{y})^2)^{-1}\frac{\partial \log p_{\boldsymbol{y}}}{\partial y_i}(\boldsymbol{y})$$

for $i = 1, \ldots, n$. The estimator verifies the constraint $\mathbb{E}_{y_i} \frac{\partial f}{\partial y_i}(\boldsymbol{y}) = 0$ for all $i = 1, \ldots, n$, and thus is still more flexible than cross-validation, as the gradients are zero only in expectation.

Another important family is related to spatially correlated noise, which is generally modeled as

$$\boldsymbol{y} = \boldsymbol{x} + \boldsymbol{\sigma} * \boldsymbol{\epsilon}$$

where $*$ denotes the convolution operator, $\boldsymbol{\sigma} \in \mathbb{R}^p$ is a vector-valued and $\boldsymbol{\epsilon} \sim \mathcal{N}(\boldsymbol{0}, \boldsymbol{I})$. If we don't know the exact noise correlation, we can consider the set of covariances with correlations up to $\pm r$ taps/pixels[3], we can minimize

$$\min_{f \in \mathcal{S}_C} \mathbb{E}_{\boldsymbol{y}} \|f(\boldsymbol{y}) - \boldsymbol{y}\|^2 \tag{16}$$

with $\mathcal{S}_C = \{f \in \mathcal{L}^1 : \mathbb{E}_{\boldsymbol{y}} \sum_{i=1}^n \frac{\partial f_i}{\partial y_{(i+j) \bmod n}}(\boldsymbol{y}) = 0, \ j \in \{-r, \ldots, r\}\}$, or equivalently

$$\min_f \max_{\boldsymbol{\eta}} \mathbb{E}_{\boldsymbol{y}} \|f(\boldsymbol{y}) - \boldsymbol{y}\|^2 + \sum_{j=-r}^r \eta_j \sum_{i=1}^n \frac{\partial f_i}{\partial y_{(i+j) \bmod n}}(\boldsymbol{y}) \tag{C-UNSURE}$$

whose solution is $f(\boldsymbol{y}) = \boldsymbol{y} + \hat{\boldsymbol{\eta}} * \nabla \log p_{\boldsymbol{y}}(\boldsymbol{y})$ where the optimal multipliers are given by $\hat{\boldsymbol{\eta}} = \boldsymbol{F}^{-1}(\boldsymbol{1}/\boldsymbol{F}\boldsymbol{h})$ where the division is performed elementwise, $\boldsymbol{h} \in \mathbb{R}^{2r+1}$ is the $\pm r$ tap autocorrelation of the score and $\boldsymbol{F} \in \mathbb{C}^{(2r+1) \times (2r+1)}$ is the discrete Fourier transform (see Appendix A for more details).

**Poisson-Gaussian noise** In many applications, the noise has a multiplicative nature due to the discrete nature of photon-counting detectors. The Poisson-Gaussian noise model is stated as (Le Montagner et al., 2014)

$$\boldsymbol{y} = \gamma \mathcal{P}(\frac{\boldsymbol{x}}{\gamma}) + \sigma \boldsymbol{\epsilon} \tag{17}$$

where $\mathcal{P}$ denotes the Poisson distribution, and the variance of the noise is dependent on the signal level, since $\mathbb{V}\{y_i | x_i\} = x_i \gamma + \sigma^2$. In this case, we minimize

$$\min_{f \in \mathcal{S}_{PG}} \mathbb{E}_{\boldsymbol{y}} \|f(\boldsymbol{y}) - \boldsymbol{y}\|^2 \tag{18}$$

with $\mathcal{S}_{PG} = \{f \in \mathcal{L}^1 : \mathbb{E}_{\boldsymbol{y}} \operatorname{div} f(\boldsymbol{y}) = 0, \ \mathbb{E}_{\boldsymbol{y}} \boldsymbol{y}^\top \nabla f(\boldsymbol{y}) = 0\}$, or equivalently

$$\min_f \max_{\eta, \gamma} \mathbb{E}_{\boldsymbol{y}} \left[ \|f(\boldsymbol{y}) - \boldsymbol{y}\|^2 + \sum_{i=1}^n 2(\eta + y_i \gamma) \frac{\partial f_i}{\partial y_i}(\boldsymbol{y}) \right] \tag{PG-UNSURE}$$

whose solution is $f(\boldsymbol{y}) = \boldsymbol{y} + (\boldsymbol{1}\hat{\eta} + \boldsymbol{y}\hat{\gamma}) \circ \nabla \log p_{\boldsymbol{y}}(\boldsymbol{y}) + \boldsymbol{1}\hat{\gamma}$ where $\hat{\eta}$ and $\hat{\gamma}$ are included in Appendix C.

**Exponential family noise distributions** We can generalize UNSURE to other noise distributions with unknown variance by considering a generalization of Stein's lemma, introduced by Hudson (1978). In this case, we consider the constrained set $\{f \in \mathcal{L}^1 : \mathbb{E}_{\boldsymbol{y}} \sum_{i=1}^n a(y_i) \frac{\partial f_i}{\partial y_i}(\boldsymbol{y}) = 0\}$ for some function $a : \mathbb{R} \mapsto \mathbb{R}$ which depends on the noise distribution. For example, the isotropic Gaussian noise case is recovered with $a(y_i) = 1$. See Appendix B for more details.

**General inverse problems** We consider problems beyond denoising $\boldsymbol{y} = \boldsymbol{A}\boldsymbol{x} + \boldsymbol{\epsilon}$, where $\boldsymbol{y}, \boldsymbol{\epsilon} \in \mathbb{R}^m$, $\boldsymbol{x} \in \mathbb{R}^n$ and $\boldsymbol{A} \in \mathbb{R}^{m \times n}$ where generally we have fewer measurements than pixels, ie. $m < n$. We can adapt UNSURE to learn in the range space of $\boldsymbol{A}^\top$ via

$$\min_f \max_\eta \mathbb{E}_{\boldsymbol{y}} \|\boldsymbol{A}^\dagger (\boldsymbol{A}f(\boldsymbol{y}) - \boldsymbol{y})\|^2 + 2\eta \operatorname{div} \left[ (\boldsymbol{A}^\dagger)^\top \boldsymbol{A}^\dagger \boldsymbol{A}f \right](\boldsymbol{y}) \tag{General UNSURE}$$

where $\boldsymbol{A}^\dagger \in \mathbb{R}^{n \times m}$ is the linear pseudoinverse of $\boldsymbol{A}$ (Eldar, 2009), or a stable approximation thereof. As with SURE, the proposed loss only provides an estimate of the error in the nullspace of $\boldsymbol{A}$. To learn in the nullspace of $\boldsymbol{A}$, we use (General UNSURE) with the equivariant imaging (EI) loss (Chen et al., 2021), which leverages the invariance of natural image distributions to geometrical transformations (shifts, rotations, etc.), please see Appendix F for more details. A theoretical analysis about learning in incomplete problems can be found in Tachella et al. (2023a).

---

[3]Here we consider 1-dimensional signals for simplicity but the result extends trivially to the 2-dimensional case.

## 4 METHOD

We propose two alternatives for learning the optimal ZED estimators: the first solves the Lagrange problem in (14), whereas the second one approximates the score during training, and uses the formula in Theorem 3 for inference.

**UNSURE** We can solve the optimization problem in (14) by parametrizing the estimator $f_{\boldsymbol{\theta}}$ using a deep neural network with weights $\boldsymbol{\theta} \in \mathbb{R}^p$, and approximating the expectation over $\boldsymbol{y}$ by a sum over a dataset of noisy images $\{\boldsymbol{y}_j\}_{j=1}^N$. During training, we search for a saddle point of the loss on $\boldsymbol{\theta}$ and $\boldsymbol{\eta}$ by alternating between a gradient descent step on $\boldsymbol{\theta}$ and a gradient ascent step on $\boldsymbol{\eta}$. The gradient with respect to $\boldsymbol{\eta}$ does not require additional backpropagation through $f_{\boldsymbol{\theta}}$ and thus adds a minimal computational overhead. We approximate the divergence using a Monte Carlo approximation (Ramani et al., 2008)

$$\mathrm{tr}\left(\boldsymbol{M} \frac{\partial f_{\boldsymbol{\theta}}}{\partial \boldsymbol{y}}(\boldsymbol{y})\right) \approx \frac{(\boldsymbol{M}\,\boldsymbol{b})^{\top}}{\tau}\left(f_{\boldsymbol{\theta}}(\boldsymbol{y} + \tau \boldsymbol{b}) - f_{\boldsymbol{\theta}}(\boldsymbol{y})\right) \tag{19}$$

where $\boldsymbol{M} \in \mathbb{R}^{n \times n}$, $\boldsymbol{b} \sim \mathcal{N}(0, I)$ and $\tau = 0.01$ is a small constant, and for computing the divergence in (UNSURE) we choose $\boldsymbol{M} = \boldsymbol{I}$, and for computing the second term in (PG-UNSURE) we choose $\boldsymbol{M} = \mathrm{diag}(\eta \boldsymbol{1} + \gamma \boldsymbol{y})$. A pseudocode for computing the loss is presented in Appendix F.

**UNSURE via score** Alternatively, we can approximate the score of the measurement data with a deep network $s_{\theta}(\boldsymbol{y}) \approx \nabla \log p_{\boldsymbol{y}}(\boldsymbol{y})$ using the AR-DAE loss proposed in Lim et al. (2020); Kim & Ye (2021),

$$\arg\min_{\boldsymbol{\theta}} \mathbb{E}_{(\boldsymbol{y},\boldsymbol{b},\tau)} \|\boldsymbol{b} + \tau s_{\boldsymbol{\theta}}(\boldsymbol{y} + \tau \boldsymbol{b})\|^2 \tag{20}$$

where $\boldsymbol{b} \sim \mathcal{N}(\boldsymbol{0}, \boldsymbol{I})$ and $\tau \sim \mathcal{N}(0, \delta^2)$. Following Lim et al. (2020), we start with a large standard deviation by setting $\delta_{\max}$ and anneal it to a smaller value $\delta_{\min}$ during training[4]. At inference time, we reconstruct the measurements $\boldsymbol{y}$ using Theorem 3, as it only requires access to the score:

$$f_{\boldsymbol{\theta}}(\boldsymbol{y}) = \boldsymbol{y} + \boldsymbol{\Sigma}_{\hat{\boldsymbol{\eta}}}\, s_{\boldsymbol{\theta}}(\boldsymbol{y})$$

where $\boldsymbol{\Sigma}_{\hat{\boldsymbol{\eta}}}$ is computed using Theorem 3 with $\boldsymbol{H} \approx \mathbb{E}_{\boldsymbol{y}}\, s_{\boldsymbol{\theta}}(\boldsymbol{y}) s_{\boldsymbol{\theta}}(\boldsymbol{y})^{\top}$. The expectation can be computed over the whole dataset or a single noisy image if the noise varies across the dataset. As we will see in the following section, this approach requires a single model evaluation per gradient descent step, compared to two evaluations when learning $f_{\boldsymbol{\theta}}$ for computing (19), but it requires more epochs to converge and obtained slightly worse reconstruction results in our experiments.

## 5 EXPERIMENTS

We show the performance of the proposed loss in various inverse problems and compare it with state-of-the-art self-supervised methods. All our experiments are performed using the deep inverse library (Tachella et al., 2023b). We use the AdamW optimizer for optimizing network weights $\boldsymbol{\theta}$ with step size $5 \times 10^{-4}$ and default momentum parameters and set $\alpha = 0.01$, $\mu = 0.9$ and $\tau = 0.01$ for computing the UNSURE loss in Algorithm 1. Examples of reconstructed images are shown in Figure 4. Results on real Cryo-EM data (Bepler et al., 2020) are included in Appendix E.

**Gaussian denoising on MNIST** We evaluate the proposed loss for different noise levels $\sigma \in \{0.05, 0.1, 0.2, 0.3, 0.4, 0.5\}$ of Gaussian noise on the MNIST dataset. We compare with Noise2Score (Kim et al., 2022), Neighbor2Neighbor (Huang et al., 2021), recorrupted2recorrupted (Pang et al., 2021) and SURE (Metzler et al., 2020) and use the same U-Net architecture for all experiments (see Appendix F for more details). The results are shown in Figure 3. The Lagrange multiplier $\eta$ converges in a few epochs to a slightly larger value of $\sigma$ for all evaluated noise levels. SURE and R2R are highly sensitive to the choice of $\sigma$, providing large errors when the noise level is wrongly specified. The proposed UNSURE loss and UNSURE via score do not require knowledge about the noise level and perform only slightly worse than the supervised case. We also evaluate the UNSURE via score approach using image-wise noise level estimations, obtaining a similar performance than averaging over the whole dataset. As the best results are obtained by the UNSURE loss, we use this variant in the rest of our experiments.

---

[4]In our Gaussian denoising experiments, we set $\delta_{\max} = 0.1$ and $\delta_{\min} = 0.001$ as in Kim et al. (2022).

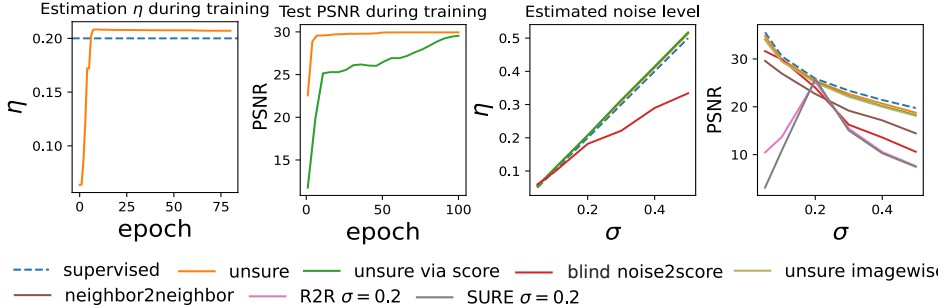

Figure 3: **MNIST denoising experiments.** From left to right: i) evolution of the Lagrange multiplier $\eta$ for the case with $\sigma = 0.2$, ii) test PSNR during training for UNSURE and UNSURE via score with $\sigma = 0.1$, iii) estimated Lagrange multiplier as a function of $\sigma$, iv) average test PSNR for various methods and various noise levels $\sigma$.

**Colored Gaussian noise on DIV2K**  We evaluate the performance of the proposed method on correlated noise on the DIV2K dataset (Zhang et al., 2017), by adding Gaussian noise convolved with a box blur kernel of $3 \times 3$ pixels. To capture the spatial structure of noise, we parameterize $\Sigma_{\eta}$ in (14) as a circulant (convolution) matrix, as explained in Section 3.1. It is worth noting that only a bound on the support of the blur is needed for the proposed method, while the specific variance in each direction is not necessary. We train all the models on 900 noisy patches of $128 \times 128$ pixels extracted from the training set and test on the full validation set which contains images of more than $512 \times 512$ pixels.

Table 1 shows the test results for the different evaluated methods (Neighbor2Neighbor, Noise2Void and SURE). The proposed method significantly outperforms other self-supervised baselines which fail to capture correlation, performing $\approx 1$ dB below the SURE method *with known* noise covariance and supervised learning. Table 2 shows the impact of the choice of the kernel size in the proposed method, where overspecifying the kernel size provides a significant improvement over underspeci-fication, and a mild performance reduction compared to choosing the exact kernel size.

| Method | Noise2Void | Neighbor2Neighbor | UNSURE (unknown $\Sigma$) | SURE (known $\Sigma$) | Supervised |
|---|---|---|---|---|---|
| PSNR [dB] | $19.09 \pm 1.79$ | $23.61 \pm 0.13$ | $28.72 \pm 1.03$ | $29.77 \pm 1.22$ | $29.91 \pm 1.26$ |

Table 1: Average test PSNR of learning methods for correlated Gaussian noise.

| Kernel size $\eta$ | $1 \times 1$ | $3 \times 3$ | $5 \times 5$ |
|---|---|---|---|
| PSNR [dB] | $23.62$ | $28.72 \pm 1.03$ | $27.38 \pm 0.88$ |

Table 2: Average test PSNR of the proposed method as a function of the chosen blur kernel size, where the ground-truth kernel had size $3 \times 3$.

**Computed tomography with Poisson-Gaussian noise on LIDC**  We evaluate a tomography problem where (resized) images of $128 \times 128$ pixels taken from the LIDC dataset, are measured using a parallel beam tomography operator with 128 equispaced angles (thus $A$ does not have a significant nullspace). The measurements are corrupted by Poisson-Gaussian noise distribution in (17) of levels $\gamma = \sigma = 0.005$. We compare our method with the Poisson-Gaussian SURE loss proposed in (Le Montagner et al., 2014), and a cross-validation approach similar to Noise2Inverse (Hendriksen et al., 2020). In this case, we use the loss in (PG-UNSURE) with $\eta$ and $\gamma$ as Lagrange multipliers, together with the pseudoinverse correction in (General UNSURE). We evaluate all methods using an unrolled network with a U-Net denoiser backbone (see Appendix F for details). Table 3 presents the test PSNR of the compared methods. The proposed loss outperforms the cross-validation approach while being close to SURE with known noise levels.

**Accelerated magnetic resonance imaging with FastMRI**  We evaluate a single-coil $2\times$-accelerated MRI problem using a subset of 900 images of the FastMRI dataset for training and

| Method | Noise2Inverse | PG-UNSURE (unknown $\sigma$, $\gamma$) | PG-SURE (known $\sigma$, $\gamma$) | Supervised |
|---|---|---|---|---|
| PSNR [dB] | $32.54 \pm 0.71$ | $33.31 \pm 0.57$ | $33.76 \pm 0.61$ | $34.67 \pm 0.68$ |

Table 3: Average test PSNR of learning methods for computed tomography with Poisson-Gaussian noise.

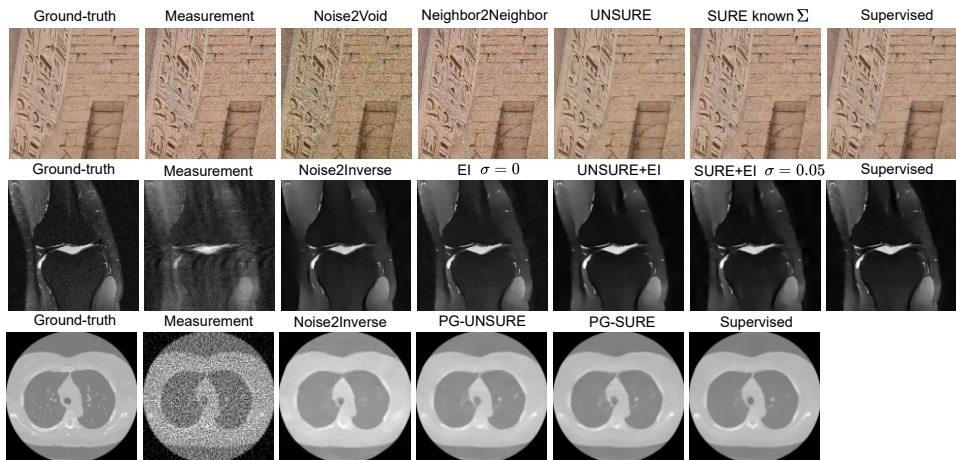

Figure 4: **Image reconstruction results for various imaging problems.** Top: colored Gaussian noise on DIV2K. Middle: Accelerated magnetic resonance imaging with FastMRI. Bottom: computed tomography with Poisson-Gaussian noise on LIDC

100 for testing (Chen et al., 2021), and adding Gaussian noise with $\sigma = 0.03$ to the k-space measurements. We evaluate all methods using an unrolled network with a U-Net denoiser backbone (see appendix for more details). We compare with cross-validation, simple measurement consistency ($\|f(\boldsymbol{y}) - \boldsymbol{y}\|^2$) and SURE with slightly wrong noise level $\sigma = 0.05$. The cross-validation method is equivalent to the SSDU method (Yaman et al., 2020). We add the EI loss with rotations as proposed in (Chen et al., 2021) to all methods since this problem has non-trivial nullspace. Table 4 shows the test PSNR for the compared methods. The proposed method performs better than all other self-supervised methods, and around 1 dB worse than supervised learning.

| Method | CV + EI | EI (assumes $\sigma = 0$) | UNSURE + EI (unknown $\sigma$) | SURE + EI (assumes $\sigma = 0.05$) | Supervised |
|---|---|---|---|---|---|
| PSNR [dB] | $33.25 \pm 1.14$ | $34.32 \pm 0.91$ | $35.73 \pm 1.45$ | $28.05 \pm 4.73$ | $36.63 \pm 1.38$ |

Table 4: Average test PSNR of learning methods for accelerated MRI with noise level $\sigma = 0.03$.

# 6 LIMITATIONS

The analysis in this paper is restricted to the $\ell_2$ loss, which is linked to minimum squared error distortion. In certain applications, more perceptual reconstructions might be desired (Blau & Michaeli, 2018), and we do not address this goal here. Both SURE and the proposed UNSURE method require an additional evaluation of the estimator during training than supervised learning, and thus the training is more computationally intensive than supervised learning.

# 7 CONCLUSION

We present a new framework to understand the robustness-expressivity trade-off of self-supervised learning methods, which characterizes different families of estimators according to the constraints in their (expected) derivatives. This analysis results in a new family of estimators which we call UNSURE, which do not require prior knowledge about the noise level, and are more expressive than cross-validation methods.

ACKNOWLEDGMENTS

Julián Tachella is supported by the ANR grant UNLIP (ANR-23-CE23-0013). Part of this work is funded by the Belgian F.R.S.-FNRS (PDR project QuadSense; T.0160.24). The authors would like to acknowledge ENS Lyon for supporting the research visits of Mike Davies and Laurent Jacques.

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

## A  PROOFS

We start by proving Theorem 3, as Propositions 1 and 2 can be then derived as simple corollaries.

*Proof.* We want to find the solution of the max-min problem

$$\min_{f} \max_{\boldsymbol{\eta}} \ \mathbb{E}_{\boldsymbol{y}} \left[ \|f(\boldsymbol{y}) - \boldsymbol{y}\|^2 + 2 \ \mathrm{tr} \left( \boldsymbol{\Sigma_\eta} \frac{\partial f}{\partial \boldsymbol{y}}(\boldsymbol{y}) \right) \right]$$

Since the problem is convex with respect to $f$ and has affine equality constraints, it verifies strong duality (Luenberger, 1969, page 236, Problem 7, Chapter 8), and thus we can rewrite it as a max-min problem, that is

$$\max_{\boldsymbol{\eta}} \min_{f} \ \mathbb{E}_{\boldsymbol{y}} \left[ \|f(\boldsymbol{y}) - \boldsymbol{y}\|^2 + 2 \ \mathrm{tr} \left( \boldsymbol{\Sigma_\eta} \frac{\partial f}{\partial \boldsymbol{y}}(\boldsymbol{y}) \right) \right].$$

Choosing $f(\boldsymbol{y}) = \boldsymbol{y} + \tilde{f}(\boldsymbol{y})$, we can rewrite the problem as

$$\max_{\boldsymbol{\eta}} \min_{\tilde{f}} \ \mathbb{E}_{\boldsymbol{y}} \|\tilde{f}(\boldsymbol{y})\|^2 + 2 \, \mathrm{tr} \left( \boldsymbol{\Sigma_\eta} \frac{\partial \tilde{f}}{\partial \boldsymbol{y}}(\boldsymbol{y}) \right) + 2 \, \mathrm{tr} \left( \boldsymbol{\Sigma_\eta} \right). \tag{21}$$

Using integration by parts and the facts that i) $p_{\boldsymbol{y}} = p_{\boldsymbol{x}} * \mathcal{N}(0, \boldsymbol{\Sigma})$ is differentiable and supported on $\mathbb{R}^n$, and ii) $\frac{\partial p_{\boldsymbol{y}}}{\partial y_i} / p_{\boldsymbol{y}} = \frac{\partial \log p_{\boldsymbol{y}}}{\partial y_i}$, we have that

$$\mathbb{E}_{\boldsymbol{y}} \frac{\partial \tilde{f}_i(\boldsymbol{y})}{\partial y_j} = \int p_{\boldsymbol{y}}(\boldsymbol{y}) \frac{\partial \tilde{f}_i}{\partial y_j}(\boldsymbol{y}) \, d\boldsymbol{y} \tag{22}$$

$$= -\int \tilde{f}_i(\boldsymbol{y}) \frac{\partial p_{\boldsymbol{y}}}{\partial y_j}(\boldsymbol{y}) \, d\boldsymbol{y} \tag{23}$$

$$= -\int \tilde{f}_i(\boldsymbol{y}) \frac{\frac{\partial p_{\boldsymbol{y}}}{\partial y_j}(\boldsymbol{y})}{p_{\boldsymbol{y}}(\boldsymbol{y})} p_{\boldsymbol{y}}(\boldsymbol{y}) \, d\boldsymbol{y} \tag{24}$$

$$= -\mathbb{E}_{\boldsymbol{y}} \, \tilde{f}_i(\boldsymbol{y}) \frac{\partial \log p_{\boldsymbol{y}}}{\partial y_j}(\boldsymbol{y}). \tag{25}$$

and thus $\mathbb{E}_{\boldsymbol{y}} \frac{\partial \tilde{f}}{\partial \boldsymbol{y}}(\boldsymbol{y}) = \mathbb{E}_{\boldsymbol{y}} \tilde{f}(\boldsymbol{y}) \nabla \log p_{\boldsymbol{y}}(\boldsymbol{y})^\top$. Plugging this result into (21), we obtain

$$\max_{\boldsymbol{\eta}} \min_{\tilde{f}_1, \dots, \tilde{f}_n} \mathbb{E}_{\boldsymbol{y}} \left[ \|\tilde{f}(\boldsymbol{y})\|^2 - 2 \operatorname{tr} \left( \boldsymbol{\Sigma}_{\boldsymbol{\eta}} \, \tilde{f}(\boldsymbol{y}) \nabla \log p_{\boldsymbol{y}}(\boldsymbol{y})^\top \right) \right] + 2 \operatorname{tr} (\boldsymbol{\Sigma}_{\boldsymbol{\eta}}) \tag{26}$$

$$\max_{\boldsymbol{\eta}} \min_{\tilde{f}_1, \dots, \tilde{f}_n} \mathbb{E}_{\boldsymbol{y}} \left[ \|\tilde{f}(\boldsymbol{y})\|^2 - 2 \tilde{f}(\boldsymbol{y})^\top \boldsymbol{\Sigma}_{\boldsymbol{\eta}}^\top \nabla \log p_{\boldsymbol{y}}(\boldsymbol{y}) \right] + 2 \operatorname{tr} (\boldsymbol{\Sigma}_{\boldsymbol{\eta}}) \tag{27}$$

where the first term can be rewritten in a quadratic form as

$$\max_{\boldsymbol{\eta}} \min_{\tilde{f}} \mathbb{E}_{\boldsymbol{y}} \|\tilde{f}(\boldsymbol{y}) - \boldsymbol{\Sigma}_{\boldsymbol{\eta}} \nabla \log p_{\boldsymbol{y}}(\boldsymbol{y})\|^2 - \operatorname{tr} \left( \boldsymbol{\Sigma}_{\boldsymbol{\eta}} H \boldsymbol{\Sigma}_{\boldsymbol{\eta}}^\top \right) + 2 \operatorname{tr} (\boldsymbol{\Sigma}_{\boldsymbol{\eta}}) \tag{28}$$

with $\boldsymbol{H} = \mathbb{E}_{\boldsymbol{y}} \nabla \log p_{\boldsymbol{y}}(\boldsymbol{y}) \nabla \log p_{\boldsymbol{y}}(\boldsymbol{y})^\top$. Since the density $p_{\boldsymbol{y}}(\boldsymbol{y}) > 0$ for all $\boldsymbol{y} \in \mathbb{R}^n$ (as it is the convolution of $p_{\boldsymbol{x}}$ with a Gaussian density), the minimum of (28) with respect to $\tilde{f}$ is

$$\tilde{f}(\boldsymbol{y}) = \boldsymbol{\Sigma}_{\boldsymbol{\eta}} \nabla \log p_{\boldsymbol{y}}(\boldsymbol{y}). \tag{29}$$

Replacing this solution in (28), we get

$$\max_{\boldsymbol{\eta}} - \operatorname{tr} \left( \boldsymbol{\Sigma}_{\boldsymbol{\eta}} H \boldsymbol{\Sigma}_{\boldsymbol{\eta}}^\top \right) + 2 \operatorname{tr} (\boldsymbol{\Sigma}_{\boldsymbol{\eta}})$$

which can be rewritten as

$$\min_{\boldsymbol{\eta}} \operatorname{tr} \left( \boldsymbol{\Sigma}_{\boldsymbol{\eta}} H \boldsymbol{\Sigma}_{\boldsymbol{\eta}}^\top \right) - 2 \operatorname{tr} (\boldsymbol{\Sigma}_{\boldsymbol{\eta}}) \tag{30}$$

In particular, if we have $\boldsymbol{\Sigma}_{\boldsymbol{\eta}} = \sum_{j=0}^{s} \eta_j \boldsymbol{\Psi}_j$ for some matrices $\{\boldsymbol{\Psi}_j \in \mathbb{R}^{n \times n}\}_{j=1}^{s}$, then the problem simplifies to

$$\min_{\boldsymbol{\eta}} \sum_{i,j} \eta_i \eta_j \operatorname{tr} \left( \boldsymbol{\Psi}_i H \boldsymbol{\Psi}_j^\top \right) - 2 \eta_i \operatorname{tr} (\boldsymbol{\Psi}_i) .$$

Setting the gradient with respect to $\boldsymbol{\eta}$ to zero results in the quadratic problem $\boldsymbol{Q} \hat{\boldsymbol{\eta}} = \boldsymbol{v}$ where $Q_{i,j} := \operatorname{tr} \left( \boldsymbol{\Psi}_i H \boldsymbol{\Psi}_j^\top \right)$ and $v_i := \operatorname{tr} (\boldsymbol{\Psi}_i)$ for $i, j = 1, \dots, s$. The matrix $\boldsymbol{Q}$ can be seen as Gram matrix since its entries can be written as inner products $\langle \boldsymbol{\Psi}_i, \boldsymbol{\Psi}_j \rangle_{\boldsymbol{H}}$ where $\boldsymbol{H}$ is positive definite due to the assumption that $p_{\boldsymbol{y}} = p_{\boldsymbol{x}} * \mathcal{N}(\boldsymbol{0}, \boldsymbol{\Sigma})$. Thus, since by assumption the set $\{\boldsymbol{\Psi}_j\}_{j=1}^{s}$ is linearly independent, $\boldsymbol{Q}$ is invertible and the optimal $\hat{\boldsymbol{\eta}}$ is given by

$$\hat{\boldsymbol{\eta}} = \boldsymbol{Q}^{-1} \boldsymbol{v}. \tag{31}$$

$\square$

We now analyze the mean squared error of ZED estimators: We can replace the solution $f(\boldsymbol{y}) = \boldsymbol{y} + \boldsymbol{\Sigma}_{\boldsymbol{\eta}} \nabla \log p_{\boldsymbol{y}}(\boldsymbol{y})$ in SURE's formula to obtain

$$\mathbb{E}_{\boldsymbol{x}, \boldsymbol{y}} \|f(\boldsymbol{y}) - \boldsymbol{x}\|^2 = \mathbb{E}_{\boldsymbol{y}} \|\boldsymbol{\Sigma}_{\boldsymbol{\eta}} \nabla \log p_{\boldsymbol{y}}(\boldsymbol{y})\|^2 + 2 \operatorname{tr} \left( \boldsymbol{\Sigma} (I + \boldsymbol{\Sigma}_{\boldsymbol{\eta}} \frac{\partial^2 \log p_{\boldsymbol{y}}}{\partial \boldsymbol{y}^2}(\boldsymbol{y})) \right) - \operatorname{tr} (\boldsymbol{\Sigma})$$

which can be written as

$$\mathbb{E}_{\boldsymbol{x}, \boldsymbol{y}} \|f(\boldsymbol{y}) - \boldsymbol{x}\|^2 = \operatorname{tr} \left( \boldsymbol{\Sigma}_{\boldsymbol{\eta}} \boldsymbol{\Sigma}_{\boldsymbol{\eta}}^\top \mathbb{E}_{\boldsymbol{y}} \nabla \log p_{\boldsymbol{y}}(\boldsymbol{y}) \nabla \log p_{\boldsymbol{y}}(\boldsymbol{y})^\top \right) + 2 \operatorname{tr} \left( \boldsymbol{\Sigma} (I + \boldsymbol{\Sigma}_{\boldsymbol{\eta}} \frac{\partial^2 \log p_{\boldsymbol{y}}}{\partial \boldsymbol{y}^2}) \right) - \operatorname{tr} (\boldsymbol{\Sigma})$$

Using the fact that $\mathbb{E}_{\boldsymbol{y}} \frac{\partial^2 \log p_{\boldsymbol{y}}}{\partial \boldsymbol{y}^2}(\boldsymbol{y}) = -\mathbb{E}_{\boldsymbol{y}} \nabla \log p_{\boldsymbol{y}}(\boldsymbol{y}) \nabla \log p_{\boldsymbol{y}}(\boldsymbol{y})^\top = -\boldsymbol{H}$, we have

$$\mathbb{E}_{\boldsymbol{x},\boldsymbol{y}} \|f(\boldsymbol{y}) - \boldsymbol{x}\|^2 = \mathrm{tr}\left(\boldsymbol{\Sigma}_{\boldsymbol{\eta}} \boldsymbol{\Sigma}_{\boldsymbol{\eta}}^\top \boldsymbol{H}\right) - 2\,\mathrm{tr}\left(\boldsymbol{\Sigma} \boldsymbol{\Sigma}_{\boldsymbol{\eta}} \boldsymbol{H}\right) + \mathrm{tr}\left(\boldsymbol{\Sigma}\right)$$

Proof of Propositions 1 and 2:

*Proof.* In this case, we have $\boldsymbol{\Sigma}_\eta = \eta \boldsymbol{I}$, and thus (31) reduces to

$$\hat{\eta} = \frac{\mathrm{tr}\left(\boldsymbol{I}\right)}{\mathrm{tr}\left(\boldsymbol{H}\right)} = \frac{n}{\mathbb{E}_{\boldsymbol{y}}\|\nabla \log p_{\boldsymbol{y}}(\boldsymbol{y})\|^2} \tag{32}$$

and the mean squared error is given by

$$\mathbb{E}_{\boldsymbol{x},\boldsymbol{y}} \|f(\boldsymbol{y}) - \boldsymbol{x}\|^2 = \hat{\eta} - \sigma^2$$

which can be rewritten as

$$\hat{\eta} = \mathbb{E}_{\boldsymbol{x},\boldsymbol{y}} \|f(\boldsymbol{y}) - \boldsymbol{x}\|^2 + \sigma^2 \tag{33}$$

Noting that $\mathrm{MMSE} = \sigma^2 - \sigma^4 \frac{1}{n} \mathbb{E}_{\boldsymbol{y}}\|\nabla \log p_{\boldsymbol{y}}(\boldsymbol{y})\|^2 = \sigma^2 - \frac{\sigma^4}{\hat{\eta}}$, we have that

$$\hat{\eta} = \frac{\sigma^4}{\sigma^2 - \mathrm{MMSE}} \tag{34}$$

Combining (33) and (12), we have

$$\mathbb{E}_{\boldsymbol{x},\boldsymbol{y}} \|f(\boldsymbol{y}) - \boldsymbol{x}\|^2 = \frac{\sigma^4}{\sigma^2 - \mathrm{MMSE}} - \sigma^2 \tag{35}$$

$$= \sigma^2 \left(\frac{1}{1 - \frac{\mathrm{MMSE}}{\sigma^2}} - 1\right). \tag{36}$$

$\square$

**Anisotropic noise**   In this case, we have $\boldsymbol{\Sigma}_\eta = \mathrm{diag}(\boldsymbol{\eta}) = \sum_{j=1}^n \eta_j \boldsymbol{e}_j \boldsymbol{e}_j^\top$, and thus we have $\boldsymbol{Q} = \mathrm{diag}(\boldsymbol{H})$ and $\boldsymbol{v} = n\boldsymbol{1}$ in (31), giving the solution

$$\hat{\eta}_i = \frac{1}{\frac{1}{n}\mathbb{E}_{\boldsymbol{y}}[\frac{\partial \log p_{\boldsymbol{y}}}{\partial y_i}(\boldsymbol{y})]^2} \tag{37}$$

for $i = 1, \dots, n$.

**Correlated noise**   If we consider the 1-dimensional signal setting, we have $\boldsymbol{\Sigma}_\eta = \mathrm{circ}(\boldsymbol{\eta}) = \sum_{j=1}^{2r+1} \eta_j \boldsymbol{T}_{j-r}$ is a circulant matrix, where $\boldsymbol{T}_j$ is the $j$-tap shift matrix, and thus $\boldsymbol{v} = n\,\boldsymbol{e}_{r+1}$ and $\boldsymbol{Q} = n\,\mathrm{circ}(\boldsymbol{h})$ where $\boldsymbol{h} \in \mathbb{R}^{2r+1}$ is the autocorrelation of $\nabla \log p_{\boldsymbol{y}}(\boldsymbol{y})$ (considering up to $\pm j$ taps) , that is

$$h_j = \frac{1}{n} \sum_{i=1}^n \mathbb{E}_{\boldsymbol{y}} \frac{\partial \log p_{\boldsymbol{y}}}{\partial y_i} \frac{\partial \log p_{\boldsymbol{y}}}{\partial y_{(i+j-r) \bmod n}}(\boldsymbol{y})$$

for $j = 1, \dots, 2r+1$, and $\boldsymbol{e}_{r+1}$ denotes the $(r+1)$th canonical vector. Thus, we have $\hat{\boldsymbol{\eta}} = \mathrm{circ}(\boldsymbol{h})^{-1} \boldsymbol{e}_{r+1}$, or equivalently

$$\hat{\boldsymbol{\eta}} = \boldsymbol{F}^{-1}(1/\boldsymbol{F}\boldsymbol{h})$$

where the division is performed elementwise and $\boldsymbol{F} \in \mathbb{C}^{(2r+1)\times(2r+1)}$ is the discrete Fourier transform.

## B   EXPONENTIAL FAMILY

We now consider separable noise distributions $p(\boldsymbol{y}|\boldsymbol{x}) = \prod_{i=1}^n q(y_i|x_i)$, where $q(x|y)$ belongs to the exponential family

$$q(y|x) = h(y)\exp(r(x)b(y) - \psi(x))$$

where $\psi(x) = \log \int h(y)\exp(r(x)b(y))$ is the normalization function.

**Lemma 4.** *Let $\boldsymbol{y} \sim p(\boldsymbol{y}|\boldsymbol{x}) = \prod_{i=1}^{n} q(y_i|x_i)$ be a random variable with mean $\mathbb{E}_{\boldsymbol{y}|\boldsymbol{x}}\boldsymbol{y} = \boldsymbol{x}$, where the distribution q belongs to the set $\mathcal{C}$ whose definition is included in Appendix A. Then,*

$$\mathbb{E}_{\boldsymbol{y}|\boldsymbol{x}}(\boldsymbol{y} - \boldsymbol{x})^\top f(\boldsymbol{y}) = \sigma^2 \sum_{i=1}^{n} \mathbb{E}_{\boldsymbol{y}|\boldsymbol{x}} a(y_i) \frac{\partial f_i}{\partial y_i}(\boldsymbol{y}) \tag{38}$$

*where $a : \mathbb{R} \mapsto [0, \infty)$ is a non-negative scalar function and $\sigma^2 = \frac{1}{n}\mathbb{E}_{\boldsymbol{y}|\boldsymbol{x}}\|\boldsymbol{y} - \boldsymbol{x}\|^2$ is the noise variance.*

Stein's lemma can be seen as a special case of Lemma 4 by setting $a(y) = 1$, which corresponds to the case of Gaussian noise. If the noise variance $\sigma^2$ is unknown, we can minimize

$$\min_{f \in \mathcal{S}_H} \mathbb{E}_{\boldsymbol{y}} \|f(\boldsymbol{y}) - \boldsymbol{y}\|^2 \tag{39}$$

where $\mathcal{S}_H = \{f \in \mathcal{L}^1 : \sum_{i=1}^{n} \mathbb{E}_{\boldsymbol{y}} a(y_i)\frac{\partial f_i}{\partial y_i}(\boldsymbol{y}) = 0\}$. Again, we consider the Lagrange formulation of the problem:

$$\max_{\eta} \min_{f} \mathbb{E}_{\boldsymbol{y}} \left[ \|f(\boldsymbol{y}) - \boldsymbol{y}\|^2 + 2\eta \sum_{i=1}^{n} a(y_i)\frac{\partial f_i}{\partial y_i}(\boldsymbol{y}) \right] \tag{40}$$

where we only need to know $a(\cdot)$ up to a proportionality constant. The solution to this problem is given by

$$f(\boldsymbol{y}) = \boldsymbol{y} + \hat{\eta} \left( a(\boldsymbol{y}) \circ \nabla \log p_{\boldsymbol{y}}(\boldsymbol{y}) + \nabla a(\boldsymbol{y}) \right).$$

where $\hat{\eta}$ is also available in closed form (see Appendix A for details). Note that when $a(y) = 1$, (40) boils down to the isotropic Gaussian noise case (UNSURE).

Lemma 4 (Hudson, 1978) applies to a subset $\mathcal{C}$ of the exponential family where $r(x) = x$, $h(y) = \frac{\partial b}{\partial y}(y) \exp(b(y)y - \int b(u)du)$ and $\frac{\partial b}{\partial y}(y) = 1/a(y)$, with $\int b(u)du$ interpreted as an indefinite integral. In this case, we aim to minimize the following problem:

$$\max_{\eta} \min_{f} \mathbb{E}_{\boldsymbol{y}} \|\boldsymbol{y} - f(\boldsymbol{y})\|^2 + 2\eta \sum_{i=1}^{n} a(y_i)\frac{\partial f_i}{\partial y_i}(\boldsymbol{y}) \tag{41}$$

for some non-negative function $a : \mathbb{R} \mapsto [0, \infty)$. As with the Gaussian case, we can look for a solution $f(\boldsymbol{y}) = \boldsymbol{y} + \tilde{f}(\boldsymbol{y})$, that is

$$\max_{\eta} \min_{\tilde{f}} \mathbb{E}_{\boldsymbol{y}} \|\tilde{f}(\boldsymbol{y})\|^2 + 2\eta \sum_{i=1}^{n} \left( a(y_i) + a(y_i)\frac{\partial \tilde{f}_i}{\partial y_i}(\boldsymbol{y}) \right) \tag{42}$$

and follow the same steps, using the following generalization of (22):

$$\mathbb{E}_{\boldsymbol{y}}\, a(y_i)\frac{\partial \tilde{f}_i(\boldsymbol{y})}{\partial y_i} = \int p_{\boldsymbol{y}}(\boldsymbol{y}) a(y_i) \frac{\partial \tilde{f}_i}{\partial y_i}(\boldsymbol{y})\, d\boldsymbol{y} \tag{43}$$

$$= -\int \tilde{f}_i(\boldsymbol{y})[a(y_i)\frac{\partial p_{\boldsymbol{y}}}{\partial y_i}(\boldsymbol{y}) + \frac{\partial a}{\partial y_i}(y_i)p_{\boldsymbol{y}}(\boldsymbol{y})]\, d\boldsymbol{y} \tag{44}$$

$$= -\int \tilde{f}_i(\boldsymbol{y})[a(y_i)\frac{\frac{\partial p_{\boldsymbol{y}}}{\partial y_j}(\boldsymbol{y})}{p_{\boldsymbol{y}}(\boldsymbol{y})} + \frac{\partial a}{\partial y_i}(y_i)]p_{\boldsymbol{y}}(\boldsymbol{y})\, d\boldsymbol{y} \tag{45}$$

$$= -\mathbb{E}_{\boldsymbol{y}}\, \tilde{f}_i(\boldsymbol{y})[a(y_i)\frac{\partial \log p_{\boldsymbol{y}}}{\partial y_j}(\boldsymbol{y}) + \frac{\partial a}{\partial y_i}(y_i)] \tag{46}$$

to obtain the solution $\tilde{f}(\boldsymbol{y}) = \eta s(\boldsymbol{y})$ with $s(\boldsymbol{y}) = a(\boldsymbol{y}) \circ \nabla \log p_{\boldsymbol{y}}(\boldsymbol{y}) + \nabla a(\boldsymbol{y})$. Replacing the optimal $\tilde{f}$ in (41), we have the following problem for $\eta$:

$$\max_{\eta} \eta^2 \mathbb{E}_{\boldsymbol{y}} \|s(\boldsymbol{y})\|^2 + 2\eta \sum_{i=1}^{n} \mathbb{E}_{y_i} a(y_i) + 2\eta^2 \sum_{i=1}^{n} \mathbb{E}_{\boldsymbol{y}} a(y_i)\frac{\partial s_i}{\partial y_i}(\boldsymbol{y}) \tag{47}$$

which is a quadratic problem whose solution is

$$\hat{\eta} = \frac{\sum_{i=1}^{n} \mathbb{E}_{y_i} a(y_i)}{\sum_{i=1}^{n} -\mathbb{E}_{\boldsymbol{y}} s_i(\boldsymbol{y})^2 - 2\mathbb{E}_{\boldsymbol{y}} a(y_i)\frac{\partial s_i}{\partial y_i}(\boldsymbol{y})} \tag{48}$$

Thus, the minimizer of (41) is

$$f(\boldsymbol{y}) = \boldsymbol{y} + \hat{\eta} \left( a(\boldsymbol{y}) \circ \nabla \log p_{\boldsymbol{y}}(\boldsymbol{y}) + \nabla a(\boldsymbol{y}) \right).$$

## C  POISSON-GAUSSIAN NOISE

The PG-SURE estimator for the Poisson-Gaussian noise model in (17) is given by (Le Montagner et al., 2014)

$$\mathbb{E}_{\boldsymbol{y}}\left[\|f(\boldsymbol{y})\|^2 + \|\boldsymbol{y}\|^2 + \sum_{i=1}^{n}2\sigma^2\frac{\partial f_i}{\partial y_i}(\boldsymbol{y}-\boldsymbol{e}_i\gamma) - 2y_i f_i(\boldsymbol{y}-\boldsymbol{e}_i\gamma) - \gamma y_i\right] - \sigma^2 \tag{49}$$

where $\boldsymbol{e}_i$ denotes the $i$th canonical vector. Following Le Montagner et al. (2014), we use the approximation

$$f_i(\boldsymbol{y}-\boldsymbol{e}_i\gamma) \approx f_i(\boldsymbol{y}) - \gamma\frac{\partial f_i}{\partial y_i}(\boldsymbol{y})$$

and

$$\frac{\partial f_i}{\partial y_i}(\boldsymbol{y}-\boldsymbol{e}_i\gamma) \approx \frac{\partial f_i}{\partial y_i}(\boldsymbol{y}) - \gamma\frac{\partial^2 f_i}{\partial y_i{}^2}(\boldsymbol{y})$$

to obtain

$$\mathbb{E}_{\boldsymbol{y}}\left[\|f(\boldsymbol{y})-\boldsymbol{y}\|^2 + \sum_{i=1}^{n}(2\sigma^2+2y_i\gamma)\frac{\partial f_i}{\partial y_i}(\boldsymbol{y}) + 2\sigma^2\gamma\frac{\partial^2 f_i}{\partial y_i{}^2}(\boldsymbol{y}) - \gamma y_i\right] - \sigma^2 \tag{50}$$

We can obtain the PG-UNSURE estimator by replacing $(\gamma,\sigma^2)$ for the Lagrange multipliers $(\gamma,\sigma)$ to obtain

$$\max_{\eta,\gamma}\min_{f}\mathbb{E}_{\boldsymbol{y}}\left[\|f(\boldsymbol{y})-\boldsymbol{y}\|^2 + \sum_{i=1}^{n}(2\eta+2y_i\gamma)\frac{\partial f_i}{\partial y_i}(\boldsymbol{y})\right] \tag{51}$$

where we drop the second order derivative, since we observe that in practice we have $|\frac{\partial^2 f_i}{\partial y_i^2}(\boldsymbol{y})| \ll |\frac{\partial f_i}{\partial y_i}(\boldsymbol{y})|$. Note that this problem is equivalent to

$$\underset{f\in\mathcal{S}_{\text{PG}}}{\arg\min}\max_{\eta,\gamma}\mathbb{E}_{\boldsymbol{y}}\|f(\boldsymbol{y})-\boldsymbol{y}\|^2 \tag{52}$$

where $\mathcal{S}_{\text{PG}} = \{f : \mathbb{E}_{\boldsymbol{y}}\text{div}f(\boldsymbol{y}) = 0, \ \mathbb{E}_{\boldsymbol{y}}\boldsymbol{y}^\top\nabla f(\boldsymbol{y}) = 0\}$ and thus we have that $\mathcal{S}_{\text{CV}} \subset \mathcal{S}_{\text{PG}} \subset \mathcal{S}_{\text{ZED}}$.

Setting $f(\boldsymbol{y}) = \boldsymbol{y} + \tilde{f}(\boldsymbol{y})$, we can rewrite problem (51) as

$$\max_{\eta,\gamma}\underset{f}{\arg\min}\sum_{i=1}^{n}\mathbb{E}_{\boldsymbol{y}}\,\tilde{f}_i(\boldsymbol{y})^2 + (2\eta+2y_i\gamma)\frac{\partial\tilde{f}_i}{\partial y_i}(\boldsymbol{y}) + 2\eta + 2\gamma y_i \tag{53}$$

Using integration by parts as in (22), we get

$$\max_{\eta,\gamma}\underset{f}{\arg\min}\sum_{i=1}^{n}\mathbb{E}_{\boldsymbol{y}}\,\tilde{f}_i(\boldsymbol{y})^2 - 2\tilde{f}_i(\boldsymbol{y})\left((\eta+y_i\gamma)\frac{\partial\log p_{\boldsymbol{y}}}{\partial y_i}(\boldsymbol{y})+\gamma\right)^2 + 2\eta + 2\gamma y_i$$

After completing squares, we obtain

$$\max_{\eta,\gamma}\underset{f}{\arg\min}\sum_{i=1}^{n}\mathbb{E}_{\boldsymbol{y}}\left(\tilde{f}_i(\boldsymbol{y}) - (\eta+y_i\gamma)\frac{\partial\log p_{\boldsymbol{y}}}{\partial y_i}(\boldsymbol{y})-\gamma\right)^2 - \left((\eta+y_i\gamma)\frac{\partial\log p_{\boldsymbol{y}}}{\partial y_i}(\boldsymbol{y})-\gamma\right)^2 + 2\eta + 2\gamma y_i$$

thus we have

$$\tilde{f}(\boldsymbol{y}) = (\boldsymbol{1}\eta+\boldsymbol{y}\gamma)\circ\nabla\log p_{\boldsymbol{y}}(\boldsymbol{y}) + \boldsymbol{1}\gamma$$

and

$$f(\boldsymbol{y}) = \boldsymbol{y} + (\boldsymbol{1}\eta+\boldsymbol{y}\gamma)\circ\nabla\log p_{\boldsymbol{y}}(\boldsymbol{y}) + \boldsymbol{1}\gamma$$

where $(\eta,\gamma)$ are given by

$$\max_{\eta,\gamma}\sum_{i=1}^{n}\mathbb{E}_{\boldsymbol{y}} - \left(\eta\frac{\partial\log p_{\boldsymbol{y}}}{\partial y_i}(\boldsymbol{y}) + y_i\gamma\frac{\partial\log p_{\boldsymbol{y}}}{\partial y_i}(\boldsymbol{y})-\gamma\right)^2 + 2\eta + 2\gamma y_i.$$

Defining the expectations $h_{a,b} := \sum_{i=1}^n \mathbb{E}_{\boldsymbol{y}} y_i^a \frac{\partial \log p_{\boldsymbol{y}}}{\partial y_i}(\boldsymbol{y})^b$ for $a, b \in \{0, 1, 2\}$, we have

$$\max_{\eta, \gamma} -\eta^2 h_{0,2} - \gamma^2 h_{2,2} - n\gamma^2 - 2\gamma\eta h_{1,2} + 2\eta\gamma h_{0,1} + 2\gamma^2 h_{1,1} + 2n\eta + 2\gamma h_{1,0}.$$

We can take derivatives with respect to $(\eta, \gamma)$ to obtain the optimality conditions

$$\begin{cases} 0 & = -\eta h_{0,2} - \gamma h_{1,2} + \gamma h_{0,1} + n \\ 0 & = -\gamma h_{2,2} - n\gamma - \eta h_{1,2} + \eta h_{0,1} + 2\gamma h_{1,1} + h_{1,0} \end{cases} \tag{54}$$

which are equivalent to

$$\begin{cases} n & = \eta h_{0,2} + \gamma(h_{1,2} - h_{0,1}) \\ h_{1,0} & = \eta(h_{1,2} - h_{0,1}) + \gamma(h_{2,2} + n + h_{1,2} - 2h_{1,1}) \end{cases}. \tag{55}$$

The solution to this system of equations is

$$\eta = \frac{n}{h_{0,2}} - \frac{\gamma(h_{1,2} - h_{0,1})}{h_{0,2}} \tag{56}$$

where

$$\gamma = \frac{h_{1,0}h_{0,2} - n(h_{1,2} - h_{0,1})}{h_{0,2}(h_{2,2} + n - 2h_{1,1}) - (h_{1,2} - h_{0,1})^2}. \tag{57}$$

In particular, if we remove the $\gamma$-constraint ($\gamma = 0$), we obtain the UNSURE formula in Proposition 1, ie. $\eta = \frac{n}{h_{0,2}} = (\frac{1}{n}\mathbb{E}_{\boldsymbol{y}}\|\nabla \log p_{\boldsymbol{y}}(\boldsymbol{y})\|^2)^{-1}$.

## D  EXAMPLES

To gain some intuition about optimal ZED estimators and their differences with cross-validation, we consider various popular signal distributions that admit an elementwise decomposition $p_{\boldsymbol{x}}(\boldsymbol{x}) = \prod_{i=1}^n q_x(x_i)$ where $q_x$ is a one-dimensional probability distribution. Since the noise distribution is also separable, we have that $p_{\boldsymbol{y}}(\boldsymbol{y}) = \prod_{i=1}^n q_y(y_i)$ where $q_y = q_x * \mathcal{N}(0, 1)$. All estimators are applied in an element-wise fashion, since there is no correlation between entries, ie. $g(y_i) = f_i(y_i)$ for all $i = 1, \ldots, n$, where $g : \mathbb{R} \mapsto \mathbb{R}$ is a one-dimensional function. The cross-validation-based estimator cannot exploit any neighbouring information, and will just return a constant value for all inputs, ie. $g(y_i) = \mathbb{E}_{y_i}\{y_i\}$, the observed mean. In contrast, both the MMSE and ZED estimators use the input $y_i$ to improve the estimation. In particular, the optimal ZED estimator is

$$g(y_i) = y_i + \frac{\nabla \log q_y(y_i)}{\mathbb{E}_{y_i}[\nabla \log q_y(y_i)]^2}$$

where $q_y = q_{\boldsymbol{x}} * \mathcal{N}(0, 1)$. Table 5 summarizes the distributions, and Figure 5 shows the estimators for each case. In all cases, we have that the cross-validation estimator is trivial $f^{\text{CV}}(\boldsymbol{y}) = \boldsymbol{0}$ where the mean squared error is simply the signal variance, whereas the ZED estimator depends on $\boldsymbol{y}$, except for the (non-sparse) Gaussian signal distribution. Interestingly, while the MMSE estimator always has non-negative derivatives (Gribonval, 2011), the optimal ZED estimator can have a negative slope.

|          | **Two Deltas** | **Gaussian** | **Spike & Slab** |
|----------|----------------|--------------|------------------|
| $q_x$    | $\frac{1}{2}\delta_{-1} + \frac{1}{2}\delta_1$ | $\mathcal{N}(0, 1)$ | $\frac{1}{2}\mathcal{N}(0, 1) + \frac{1}{2}\delta_0$ |
| $q_y$    | $\frac{1}{2}\mathcal{N}(-1, \sigma^2) + \frac{1}{2}\mathcal{N}(1, \sigma^2)$ | $\mathcal{N}(0, 1 + \sigma^2)$ | $\frac{1}{2}\mathcal{N}(0, 1 + \sigma^2) + \frac{1}{2}\mathcal{N}(0, \sigma^2)$ |
| MMSE     | 0.017 | 0.059 | 0.043 |
| MSE CV   | 0.250 | 1 | 0.500 |
| MSE ZED  | 0.024 | 1 | 0.135 |

Table 5: Examples of popular signal distributions, and the mean squared errors (MSE) for different estimators.

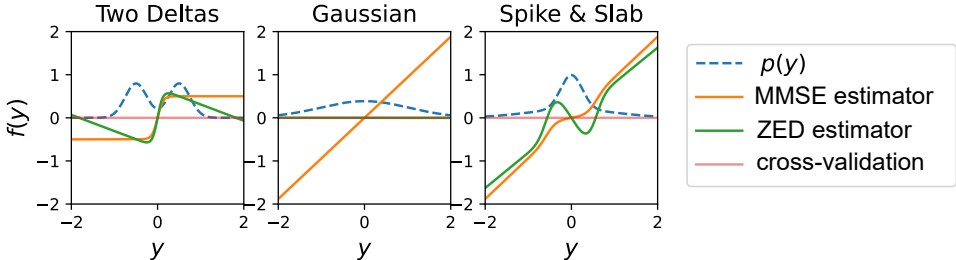

Figure 5: Comparison between MMSE, ZED, and cross-validation estimators for different toy signal distributions.

# E   REAL CRYO-EM DATA EXPERIMENT

We evaluate the proposed UNSURE method on real Cryo-EM data (provided by the Topaz-EM open-source library (Bepler et al., 2020)) whose noise distribution is unknown. The dataset consists of 5 images of $7676 \times 7420$ pixels. We applied the Poisson-Gaussian noise variant of our method (PG-UNSURE). For comparison, we included results using the Noise2Noise approach introduced by Bepler et al. (2020), which leverages videos of static images to obtain pairs of noisy images of the same underlying clean image. Both methods are evaluated using a U-Net architecture of the deep inverse library (Tachella et al., 2023b) with 4 scales and no biases. Training is done using random crops of $512 \times 512$ pixels the Adam optimizer with a learning rate of $5 \times 10^{-4}$ and standard hyperparameters. Our method receives as input the whole video averaged over time, and thus benefits from a higher signal-to-noise ratio input image than the Noise2Noise variant, whose input only averages over half of the video frames to obtain the pairs. Figure 6 shows the raw noisy image and the denoised counterparts. The proposed method produces good visual results, with fewer artifacts than the Noise2Noise approach.

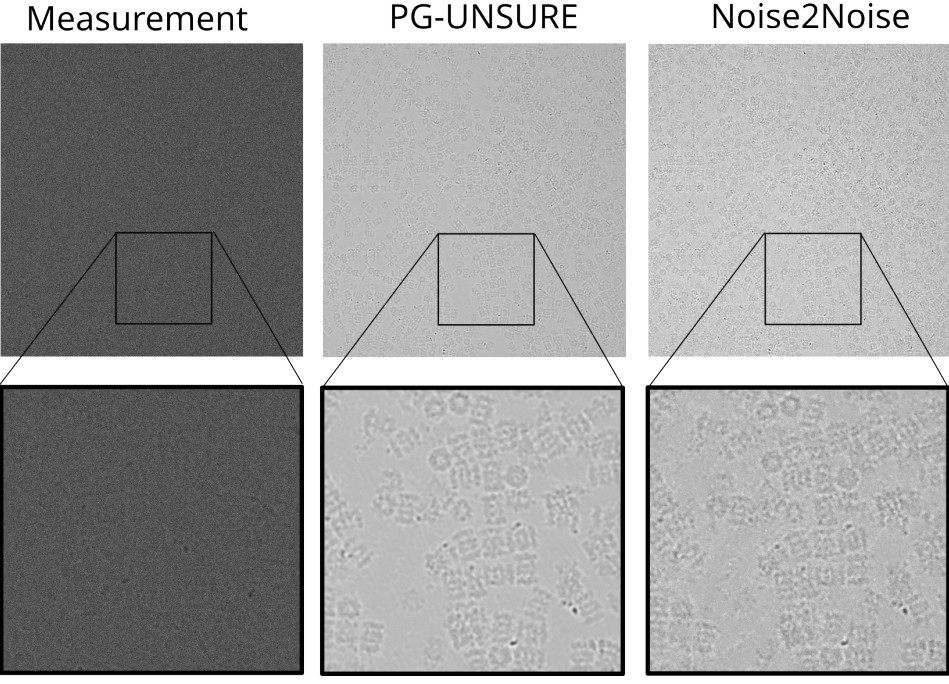

Figure 6: Blind denoising results of Cryo-EM data using the PG-UNSURE method and Noise2Noise (Bepler et al., 2020). Since the image is very large $7676 \times 7420$ pixels, is recommended to zoom in to observe the details.

# F  EXPERIMENTAL DETAILS

We use the U-Net architecture of the deep inverse library (Tachella et al., 2023b) with no biases and an overall skip-connection as a backbone network in all our experiments, only varying the number of scales of the network across experiments.

**MNIST denoising**    We use the U-Net architecture with 3 scales.

**DIV2K denoising**    We use the U-Net architecture with 4 scales.

**Computed Tomography on LIDC**    We use an unrolled proximal gradient algorithm with 4 iterations and no weight-tying across iterations. The denoiser is set as the U-Net architecture with 2 scales.

**Accelerated MRI on FastMRI**    We use an unrolled half-quadratic splitting algorithm with 7 iterations and no weight-tying across iterations. The denoiser is set as the U-Net architecture with 2 scales.

---

**Algorithm 1** UNSURE loss. The standard SURE loss is recovered by setting $\alpha = 0$, $\mathbf{\Sigma_\eta} = \mathbf{\Sigma}$.

---

**Require:** step size $\alpha$, momentum $\mu$, Monte Carlo approximation constant $\tau$

   residual $\leftarrow \|f_{\boldsymbol{\theta}}(\boldsymbol{y}) - \boldsymbol{y}\|^2$

   $\boldsymbol{b} \sim \mathcal{N}(\mathbf{0}, \boldsymbol{I})$

   div $\leftarrow \frac{(\mathbf{\Sigma_\eta}\boldsymbol{b})^\top}{\tau}\left(f_{\boldsymbol{\theta}}(\boldsymbol{y} + \tau\boldsymbol{b}) - f_{\boldsymbol{\theta}}(\boldsymbol{y})\right)$

   loss $\leftarrow$ residual $+$ div

   $\boldsymbol{g} \leftarrow \mu\boldsymbol{g} + (1 - \mu)\frac{\partial\text{div}}{\partial\boldsymbol{\eta}}$  # *average stochastic gradients w.r.t.* $\boldsymbol{\eta}$

   $\boldsymbol{\eta} \leftarrow \boldsymbol{\eta} + \alpha\boldsymbol{g}$  # *gradient ascent Lagrange multipliers*

   **return** loss($\boldsymbol{\theta}$)

---

