# OpenReview forum: "UNSURE: self-supervised learning with Unknown Noise level  and Stein's Unbiased Risk Estimate"
_ICLR.cc/2025/Conference — ICLR 2025 Poster_

### Official Review · Reviewer_K2y6 · 2024-10-29

**Soundness:** 3
**Presentation:** 2
**Contribution:** 2
**Rating:** 3
**Confidence:** 3

**Summary:**

The paper deals with the problem of designing an image/signal denoising algorithm given a sample of noisy images/signals $y_1,\ldots,y_N$ from $\mathbb R^n$. It is assumed that each $y_i$ satisfies the relation $y_i = x_i + \sigma\epsilon_i$, $i=1,\ldots,N+1$, where $\epsilon_i$ is independent of $x_i$ and is drawn from the Gaussian distribution $\mathcal N(0,\mathbf I)$. The proposed method is solve with respect to $f$ and $\eta\in\mathbb R$
the optimization problem
$$
\min_f \max_{\eta} \frac1N \sum_{i=1}^N \Big(|f(y_i) - y_i|^2 + 2\eta \ \textrm{div} f(y_i)\Big),
$$
where $f$ is chosen from a parametric class of functions, such as neural networks with a given architecture and varying weights.

Extensions to the setting of a noise covariance matrix of a more general form are also considered. In particular, if $y_i = x_i + \Sigma^{1/2}\epsilon_i$ with $\Sigma = \sum_{j=1}^s \eta_j \Psi_j$, where $\Psi_1,\ldots,\Psi_s$ are some known matrices and $\eta\in\mathbb R^s$ is an unknown vector, the proposal is to solve the saddle point problem
$$
\min_f \max_{\eta\in\mathbb R^s} \frac1N \sum_{i=1}^N \Big(|f(y_i) - y_i|^2 + 2\sum_{j=1}^s \eta_j \ \textrm{Tr} [\Psi_jf(y_i)]\Big).
$$
(There is a mistake in this formula as it is written in the paper, the correct form should be
$$
\min_f \max_{\eta\in\mathbb R^s} \frac1N \sum_{i=1}^N \Big(|f(y_i) - y_i|^2 + 2\sum_{j=1}^s \eta_j \ \textrm{Tr} \big[\Psi_j \ Df(y_i)\big]\Big)
$$
where $Df$ is the Jacobian matrix of $f$.

The main mathematical result of the paper, stated in Theorem 3, provides the explicit form of the minimizer of the problem above.

**Strengths:**

The paper is pleasant to read. The proposed method is based on a simple idea and the experiments show that it improves on some methods that have been recently proposed in the literature (such as Noise2Void, Neighbor2Neighbor).

**Weaknesses:**

1. Mathematical analysis of the proposed methodology is very light. In particular, neither the properties of the proposed estimator nor the convergence of the optimization algorithm are studied.

2. The framework for the learning problem is not clearly defined. The abstract and introduction reference supervised and self-supervised settings multiple times, but how these settings are interpreted in image reconstruction remains unclear. This only becomes explicit at the bottom of page 7, where it is clarified that the available data consist of $y_1, \ldots, y_N$. I recommend moving this clarification to the beginning of Section 2.

3. In Sections 2 and 3, the term estimator is repeatedly used to describe objects that cannot be computed directly from the data, as they rely on knowledge of the distribution of $y$. To avoid this misuse of terminology, it would be preferable to use a more accurate term.

4. Minor comments and typos:
- In the second paragraph of Section 1 on page 1, the two items starting with "i) SURE" and "ii) Noise2Self" are presented in a different order than in the abstract. For the sake of consistency throughout the paper, it would be beneficial to align their order.
- Page 3, the two lines just above Eq. (5): I guess $\mathbb R^m$ should be replaced with $\mathbb R^n$.
- Page 3, Eq. (4): This formula is easy to obtain, but it might be useful to provide a reference for non-experts.
- Page 4, Eq. (UNSURE): I suggest putting parentheses around the expression within the expectation, to underline that the divergence is also under the expectation.
- Something wrong with formula (14). The gradient of $f$ should appear in the second term.
- Page 6: "are more flexible" -> "are more flexible than"
- Page 6: "we can considering" -> "we can consider"

5. The setting under consideration closely resembles what is commonly known as the empirical Bayes framework. A discussion of the relevant literature on empirical Bayes methods would be a valuable addition.

6. The title of the paper appears somewhat misleading. It suggests that the paper will propose an unbiased estimator of the risk of an estimator $f(y)$ in the case where $\sigma$ is unknown, implying that the unbiased risk estimator would not depend on $\sigma$. However, the paper instead restricts attention to estimators $f$ for which the Stein unbiased risk estimate (SURE) happens to be independent of $\sigma$.

**Questions:**

An important argument in the proof of Theorem 3 is the use of strong duality. The authors refer to (Luenberger, 1969, Chapter 8). I believe they are referencing Theorem 1 on page 224. (If this is indeed the case, it would be helpful to highlight it in the paper.) I have a question related to the application of this theorem. Assuming that the constraint is of the form $G(x) \leqslant \theta$, the theorem requires the existence of $x_1$ such that the strict inequality $G(x_1) < \theta$ holds. My impression is that this condition cannot be satisfied in the setting under consideration. However, I may have overlooked something. Could the authors please clarify this?

---

> ### Author Response · Authors · 2024-11-19
> **Response to Reviewer K2y6 (part I)**
>
> Thanks for your comments. We believe there has been a misunderstanding regarding i) the scope of the paper which presents results not restricted to Gaussian noise, and ii) the theoretical analysis of the proposed method which provides a sharp characterization of the performance with respect to the gold standard minimum mean squared error estimator.
>
> > It is assumed that each $y_i$ satisfies the relation $y_i = x_i + \sigma\epsilon_i$, $i=1,\ldots,N+1$, where $\epsilon_i$  is independent of $x_i$ and is drawn from the Gaussian distribution $\mathcal N(\mathbf 0,\mathbf I)$.
>
> This is incorrect. The paper covers a range of scenarios including but not limited to the stated scenario. We consider Gaussian with unknown correlation (as mentioned by the reviewer), as well as Poisson and Poisson-Gaussian, and even a wider set of exponential family distributions (Appendix B). We also consider more general inverse problem scenarios in section 3.
>
> > Mathematical analysis of the proposed methodology is very light. In particular, neither the properties of the proposed estimator nor the convergence of the optimization algorithm are studied
>
> We believe there is a misunderstanding regarding this point. In the case of isotropic Gaussian noise, **the expected performance of the optimal estimator is fully characterized** in terms of the oracle MMSE estimator (Proposition 2 in the main paper). To the best of our knowledge, such a tight characterization of the performance of the optimal unsupervised estimator has not been provided for other widely popular unsupervised losses (eg. Noise2Void, Noise2Self, etc.).
>
> The analysis in the paper also shows that the loss is convex with respect to $f$ and has a global minimum (whose expression is given in closed form) if we consider the set of $f \in L_1$. For the more specific case of a neural network $f_{\theta}$, the convergence of the training algorithm is highly dependent on the specific choice of neural network architecture and is thus out of the scope of the paper.  Analyzing the convergence of general neural network training algorithms is a wide problem that remains open, and this paper is not hoping to provide a solution.
>
> > The framework for the learning problem is not clearly defined. The abstract and introduction reference supervised and self-supervised settings multiple times, but how these settings are interpreted in image reconstruction remains unclear.  This only becomes explicit at the bottom of page 7, where it is clarified that the available data consists of $y_1, \ldots, y_N$. I recommend moving this clarification to the beginning of Section 2.
>
> The statement that we are training on the noise measurements $y_1,...y_n$ is made at the bottom of the first paragraph of the introduction: "*self-supervised learning methods for image reconstruction have been proposed that can learn from noisy data alone, bypassing the need for ground-truth references*". The notion of self-supervised learning for denoising is already well established in the literature, following the large body of existing and highly cited work in this area, from Noise2Noise, Noise2self, onwards. Nonetheless, we have included in the updated manuscript a footnote making explicit that in practice the expectations over $y$ are replaced by a sum over a dataset $y_1,...y_n$.
>
> > In Sections 2 and 3, the term estimator is repeatedly used to describe objects that cannot be computed directly from the data, as they rely on knowledge of the distribution of $y$. To avoid this misuse of terminology, it would be preferable to use a more accurate term
>
> We agree that this is a common abuse of terminology, although it aligns with almost all other papers on the subject. We have modified the manuscript to better distinguish between an empirical estimator and the exact expectations (see also point above).

---

> > ### Author Response · Authors · 2024-11-19
> > **Response to Reviewer K2y6 (part II)**
> >
> > ### Minor points and questions
> >
> > > There is a mistake in this formula as it is written in the paper, the correct form should be $$ \arg \min_{\eta} \frac{1}{N} \sum_{i=1}^N \| f(y_i)-y_i\|^2 + 2 \sum_{j=1}^s \eta_j \text{tr}(\Psi_j D f(y_i))$$
> >
> > Many thanks for spotting this typo. We have corrected it in the updated manuscript.
> >
> > > In the second paragraph of Section 1 on page 1, the two items starting with "i) SURE" and "ii) Noise2Self" are presented in a different order than in the abstract. For the sake of consistency throughout the paper, it would be beneficial to align their order
> >
> > We have applied this suggestion in the updated manuscript.
> >
> > > Page 3, the two lines just above Eq. (5): I guess $\mathbb R^m$ should be replaced with $\mathbb R^n$.
> >
> > We agree with the reviewer, at this stage we are still only considering the denoising case so $m=n$. We have fixed this issue in the updated manuscript.
> >
> > > Page 3, Eq. (4): This formula is easy to obtain, but it might be useful to provide a reference for non-experts
> >
> > The derivation of this formula can be seen as a special case of our proof of Theorem 3, where the Lagrange multipliers are fixed.
> >
> > > Page 4, Eq. (UNSURE): I suggest putting parentheses around the expression within the expectation, to underline that the divergence is also under the expectation
> >
> > We agree with the reviewer and have applied this change to the updated manuscript.
> >
> > > Page 6: "are more flexible" -> "are more flexible than"
> > > Page 6: "we can considering" -> "we can consider
> >
> > Thank you for spotting these, we have applied these changes to the updated manuscript.
> >
> > > The setting under consideration closely resembles what is commonly known as the empirical Bayes framework. A discussion of the relevant literature on empirical Bayes methods would be a valuable addition
> >
> > Despite not making an explicit mention of empirical Bayes methods, the paper did include some references to empirical Bayes literature such as [Raphan & Simoncelli, 2011]. We have included some additional references ([Robbins, *The empirical Bayes approach to statistical decision problems*, 1955], [Efron, *Tweedie’s Formula and Selection Bias*, 2011], [Efron, *Large-Scale Inference: Empirical Bayes Methods for Estimation, Testing, and Prediction*, 2012]) in the updated manuscript, and explicited the link with (nonparametric) empirical Bayes estimators. It is worth noting however that to the best of our knowledge, there are no previous nonparametric empirical Bayes methods that handle partially unknown likelihood models as considered in our paper.
> >
> > > The title of the paper appears somewhat misleading. It suggests that the paper will propose an unbiased estimator of the risk of an estimator $f(y)$ in the case where $\sigma$ is unknown, implying that the unbiased risk estimator would not depend on $\sigma$. However, the paper instead restricts attention to estimators $f$ for which the Stein unbiased risk estimate (SURE) happens to be independent of $\sigma$.
> >
> > We have modified the title of the paper to "UNSURE:  self supervised learning with Unknown Noise level and Stein's Unbiased Risk Estimate" to avoid confusion.
> >
> > > An important argument in the proof of Theorem 3 is the use of strong duality. The authors refer to (Luenberger, 1969, Chapter 8). I believe they are referencing Theorem 1 on page 224. (If this is indeed the case, it would be helpful to highlight it in the paper.) I have a question related to the application of this theorem. Assuming that the constraint is of the form $G(x) \leqslant \theta$, the theorem requires the existence of $x_1$ such that the strict inequality $G(x_1) < \theta$ holds. My impression is that this condition cannot be satisfied in the setting under consideration. However, I may have overlooked something. Could the authors please clarify this?
> >
> > We use Theorem 1 of page 217 of (Luenberger, 1969, Chapter 8). Our case corresponds to a simplified setting of the theorem where $G$ represents a hard constraint $\mathbb{E}_{y} \text{div} f(y)=0$,  and thus we can define a $G: L_1 \to\mathbb{R}^2$ with
> >
> >  $G_1(f)=-\mathbb{E}_{y} \text{div} f(y)$ and
> >
> > $G_2(f)=\mathbb{E}_{y} \text{div} f(y)$.
> >
> > Thus asking for the condition  $G(f) \leqslant 0$ is simply asking that there exist an $f\in L_1$ such that  $\mathbb{E}_{y} \text{div} f(y)=0$, which is well satisfied in our setting.
> >  We have included the reference to page 217 and this short explanation in the updated manuscript.

---

> > > ### Comment · Reviewer_K2y6 · 2024-12-02
> > > **The proof is incorrect**
> > >
> > > The reply posted by the authors confirms my initial opinion that the proof has a flaw. Theorem 1 from Luenberger cannot be used as it is mentioned by the authors. This theorem explicitly requires the existence of a point denoted by $x_1$ such that $G(x_1)<0$. For $G$ defined in the authors' reply, it is obvious that this condition cannot be fulfilled.

---

> > > > ### Author Response · Authors · 2024-12-02
> > > >
> > > > We believe there is a misunderstanding here, the proof is not incorrect. In Chapter 8, page 236 of Luenberger (see problem 7) a corollary of Theorem 1 is presented for the case of a finite number of equality constraints, only requiring that the equality constraint is feasible (where the linear constraints are $H(x_1)=\theta$ in Luenberger's notation). This condition is simply that there exists an $f\in \mathcal{L}_1$ such that $\mathbb{E}_y \text{div} f(y)=0$, as $H(f)=\mathbb{E}_y \text{div} f(y)$ (which is a finite linear constraint) and $\theta=0$.
> > > >
> > > > We will make sure that this is clear in the final manuscript, dropping the $G$ notation of the footnote of page 14 for that of $H$ in problem 7 of Luenberger, page 236.

---

> > > > > ### Comment · Reviewer_K2y6 · 2024-12-02
> > > > > **Issue with the proof**
> > > > >
> > > > > You claim that you use Theorem 1 of page 217. You also wrote in the reply the expression of the function $ G$ in your case. I simply say that the mentioned theorem requires the existence of a point $x_1$ such that $G(x_1)<0$. This condition cannot be satisfied with your choice of $G$, since it would imply $h(x_1)<0$ and $h(x_1)>0$, for some function $h$, which is impossible.

---

> > > > > > ### Author Response · Authors · 2024-12-02
> > > > > >
> > > > > > The proof relies on the extension of Theorem 1 (page 217 of Luenberger) presented on page 236 of Luenberger (Problem 7) for equality constraints, which does not require $H(x_1)<0$ and $H(x_1)>0$ (which we agree with the reviewer that is impossible to verify), but only that there exist $x_1$ such that $H(x_1)=0$ (please see the details in Luenberger, page 236), which is well verified in our setting.
> > > > > >
> > > > > > We are sorry for the confusion regarding the $G$ notation in our first reply, we agree that we should have used the $H$ notation of Luenberger. As stated in our previous response, we will make sure that the final manuscript (in particular, the footnote of page 14) will refer to the $H$, page 236 of Luenberger, to avoid any confusion.

---

> > ### Comment · Reviewer_K2y6 · 2024-12-02
> > **reply to the rebuttal**
> >
> > I appreciate the answers provided by the authors, but I do not find that they addressed satisfactorily my concerns.
> >
> > Propositions 1 and 2 deal with the pseudo-estimators. Indeed $f^{ZED}$ defined by Eq (7-8) minimizes the so-called population version of a criterion, that involves the expectation with respect to an unknown distribution. Therefore, one cannot claim that Proposition 2 characterizes the risk of the estimator. The risk of an estimator, or a learning procedure, based on $n$ observations should necessarily depend on the sample size $n$ and, hopefully, go to zero as $n\to\infty$. The right-hand side of Eq (10) in Proposition 2 does not depend on $n$. This is the reason underlying my evaluation.

---

> > > ### Author Response · Authors · 2024-12-02
> > >
> > > Thanks for your feedback. The analysis in the paper (including Propositions 1 and 2) focuses on the asymptotic case $n\to\infty$, where we replace empirical sums over a dataset of $n$ noisy observations by expectations $\mathbb{E}_y$. The asymptotic setting allows us to provide sharp characterizations of the resulting zero-expected divergence estimator $f^{\text{ZED}}$, and compare its performance with the oracle MMSE estimator. An analysis of the non-asymptotic setting (finite $n$) would require further regularity assumptions on $p_y$ and $f$ which are out of the scope of the paper.
> > >
> > > The tight characterization of the performance of the optimal ZED estimator provided in Propositions 1 and 2 has not been provided for other widely popular self-supervised losses (e.g., Noise2Void, Noise2Self, etc.). Moreover, the non-asymptotic case is evaluated empirically on a wide set of numerical experiments, showing that
> > > 1. **Denoisers trained with finite $n$ perform very closely to the predictions of our asymptotic bounds** (Figure 2).
> > > 2. The proposed self-supervised objective outperforms other self-supervised methods in various imaging inverse problems (Section 5), as predicted by the theoretical analyses.

---

### Official Review · Reviewer_bm6s · 2024-11-01

**Soundness:** 3
**Presentation:** 4
**Contribution:** 3
**Rating:** 8
**Confidence:** 3

**Summary:**

This paper introduces a novel framework that bridges this gap by proposing a SURE-inspired method that can effectively learn from noisy data without explicit noise-level information. The authors provide theoretical analysis of their proposed method and through experiments, the authors demonstrate good performance compared to existing self-supervised methods on various imaging inverse problems.

**Strengths:**

This manuscript presents a well-structured and comprehensive review of existing denoising techniques. The clear and logical organization of the content enhances readability.

**Weaknesses:**

While the contribution of this manuscript may not be groundbreaking, it offers a valuable insight into denoising techniques. Regarding the title, a more descriptive title such as "Unknown Noise Level Estimation via Self-supervised Stein's Unbiased Risk Estimator" would provide a clearer indication of the paper's focus. Additionally, minor punctuation errors are present throughout the manuscript, which could be addressed in a future revision.

**Questions:**

(n/a)

---

> ### Author Response · Authors · 2024-11-19
> **Response to Reviewer bm6s**
>
> Thank you for your comments.
>
> > Regarding the title, a more descriptive title such as "Unknown Noise Level Estimation via Self-supervised Stein's Unbiased Risk Estimator" would provide a clearer indication of the paper's focus.
>
> Thank you for this suggestion. The proposed method is not using Stein's Unbiased Risk Estimator (SURE) to estimate the noise level, but rather proposing a new self-supervised loss altogether that ressembles SURE but does not require access to the noise level as SURE. The resulting estimator does not estimate explicitly the noise level, but rather provides an upper bound as a byproduct through the optimal Lagrange multiplier.
>
> We have modified the title of the paper to "UNSURE:  self-supervised learning with Unknown Noise level  and Stein's Unbiased Risk Estimate" to indicate the focus on self-supervised learning.

---

> > ### Comment · Reviewer_bm6s · 2024-11-24
> >
> > Thank you for your response and addressing my concern. I will maintain my original rating as it reflects my overall assessment of the manuscript.

---

### Official Review · Reviewer_W3qG · 2024-11-03

**Soundness:** 3
**Presentation:** 3
**Contribution:** 2
**Rating:** 8
**Confidence:** 4

**Summary:**

This paper addresses recent advances in self-supervised learning methods for image reconstruction that learn directly from noisy data, eliminating the need for ground-truth data. The authors introduce a theoretical framework that examines the trade-off between expressivity and robustness in noise-handling techniques.

**Strengths:**

-  The authors introduce a theoretical framework that examines the trade-off between expressivity and robustness in noise-handling techniques.

- The method does not require knowledge about the noise level to denoise images.

- Numerical validation is presented to verify the ability of the proposed methodology to estimate the noise level.

**Weaknesses:**

- A primary limitation of this paper is that the denoising experiments are carried out on synthetic scenarios i.e. a clean image is corrupted by noise.  Could the authors consider testing their denoising method and noise level estimation in a more realistic scenario? For instance Noise2Noise (method the authors claim to outperform) has been tested in experimental cryo-em images (see link below). Testing the proposed method in such a scenario would provide valuable insights into its practical applicability, particularly in terms of noise level estimation and the quality of the denoised output.

https://github.com/tbepler/topaz/blob/master/tutorial/04_denoising.ipynb

- The image dataset studied in this paper are relatively of low level of details (mostly low frequency images), raising an important question about robustness. How well would the algorithm perform when trained on images with higher levels of detail? Using the cryo-em images would be interesting scenario since resolution it is the most important factor to determine the performance of any denoising method.

**Minor Comment**

- I suggest the authors to change the notation of matrices in capital boldface to be consistent with the current notation of vectors.

**Questions:**

- On top of my previous comment, the authors did not discuss if the estimation of noise level has limitations in extreme cases? Could the authors discuss whether the noise level estimator is still accurate in low SNR cases which are also common in imaging applications?

- For the case of General inverse problems, it is interesting the proposed method is able to adapt to the range of $A^{T}$. However, it posses the question, could the authors discuss the stability of the proposed method where the dimension of the range is small? What is the limit in terms of range dimension so the propose method is able to accurately estimate the noise level?

- Based on my previous comments some discussion is needed on the performance of the proposed method when the estimation of noise level is performed in extreme cases, and low dimension range of $A^{T}$.

---

> ### Author Response · Authors · 2024-11-19
> **Response to Reviewer W3qG (part I)**
>
> Thank you for your comments. We have evaluated the proposed method on the real Cryo-EM data suggested by the reviewer in the updated manuscript, proving the *effectiveness of the method with real noisy data*.
>
> > Could the authors consider testing their denoising method and noise level estimation in a more realistic scenario? Testing the proposed method in such a scenario would provide valuable insights into its practical applicability, particularly in terms of noise level estimation and the quality of the denoised output.
>
> Adding synthetic noise allows us to present a well-grounded quantitative comparison of the performance of the different denoisers. Nonetheless, we have evaluated the Poisson-Gaussian variant of the proposed method (PG-UNSURE) in the Cryo-EM images provided by the reviewer, and added the results to the appendix of the updated manuscript. The proposed method significantly reduces the noise, and unveils the different structures present in the images. Please see Appendix E (Figure 6) of the updated manuscript for more details.
>
> > For instance Noise2Noise (method the authors claim to outperform) has been tested in experimental cryo-em images.
>
> There seems to be a misunderstanding regarding this point. We do not claim in the paper that we outperform the Noise2Noise method which requires **2 independent noise realizations** per signal in the dataset, but rather that we outperform the Noise2Void, Noise2Self and Noise2Inverse methods that use a **single noisy realization** per signal.
>
> The Noise2Noise method requires access to strictly more information (2 noisy realizations of the same signal) than the proposed method (single noise realisation). The pairs used in Noise2Noise provide significant information about the noise distribution, as they offer a histogram of $p(y|x)$ of 2 realizations per clean value $x$. Thus, we do not expect UNSURE to perform better than Noise2Noise. However, our method is more broadly applicable than Noise2Noise, as in most imaging applications we do not have access to two independent noise copies.
>
> > The image dataset studied in this paper are relatively of low level of details (mostly low frequency images), raising an important question about robustness. How well would the algorithm perform when trained on images with higher levels of detail?
>
> The images considered, such as patches from 512x512 DIV2K dataset of natural images, contain significant structure and detail, and certainly more detail than is present in cryo-EM images. Moreover, the DIV2K, LIDC and FastMRI datasets are widely used in the literature. Note that as the trained networks have convolutional architectures, it is possible to evaluate them on match larger images at test time.
>
> > I suggest the authors to change the notation of matrices in capital boldface to be consistent with the current notation of vectors.
>
> Thank you for this suggestion. We have applied the proposed change in the updated manuscript.
>
> > the authors did not discuss if the estimation of noise level has limitations in extreme cases? Could the authors discuss whether the noise level estimator is still accurate in low SNR cases which are also common in imaging applications
>
> There might be a confusion regarding noise level estimation: the proposed method *does not* explicitly the noise level $\sigma^2$, but rather obtains the upper bound $\hat{\eta}$ as a byproduct of the Lagrangian optimization. The output of the learning algorithm is not a noise level, but rather a blind denoiser.
>
> Moreover, the proposed method obtains very good empirical performances in the low SNR regime:
>
> 1. The results in Figure 3 show as $\sigma^2$ grows, the gap between the standard denoiser and its ZED variant becomes smaller.
> 2. the real cryo-EM data experiments in the updated manuscript which suffer from very low SNR.
>
> Finally, the good performance at low SNR can be understood by the theory provided in the paper: Proposition 2 states that the proposed method provides the following upper bound estimate $\hat{\eta}$ whose difference with the true noise variance $\sigma^2$ is: $$\hat{\eta} - \sigma^2 = \frac{\sigma^2}{1-\text{MMSE}/\sigma^2} - \sigma^2 = \text{MMSE} + \sigma^2 \sum_{j=2}^{\infty} (\frac{\text{MMSE}}{\sigma^2})^j
> $$ where $\text{MMSE}$ is the minimum mean squared error, that is the minimum error that *any* (supervised or unsupervised) denoiser can achieve.  At high-noise levels, $\text{MMSE}$ saturates, becoming independent of $\sigma^2$. To see this, consider the extreme case of $\sigma\to\infty$, where the optimal denoiser should simply return the mean of the clean signal distribution $f(y)= \mathbb{E} \, x$, fully disregarding the input $y$.  Thus, we have that $\hat{\eta} - \sigma^2 \to 0$ as $\sigma\to\infty$.
>
> Note that the expected reconstruction error also follows the same analysis $\mathbb{E}_{x,y}\|f^{\text{ZED}}(y) - x \|^2$ and thus we can expect to perform better in low SNR settings.

---

> > ### Author Response · Authors · 2024-11-19
> > **Response to Reviewer W3qG (part II)**
> >
> > > For the case of general inverse problems, it is interesting the proposed method is able to adapt to the range of $A^{\top}$. However, it posses the question, could the authors discuss the stability of the proposed method where the dimension of the range is small? What is the limit in terms of range dimension so the propose method is able to accurately estimate the noise level?
> >
> > We believe there is some confusion regarding this point. In the proposed method, the noise level is estimated across the dataset of images since there is a single Lagrange multiplier $\eta$ in eq. (UNSURE) for the whole dataset. Thus, even if there is a small number of measurements per signal (a small dimension of the range of $A^{\top}$), the method can still obtain an accurate estimation of the noise level.
> >
> > We have included this discussion regarding the high noise setting in the updated manuscript.

---

> > > ### Author Response · Authors · 2024-12-02
> > >
> > > Please let us know if you have any questions regarding our response.
> > > If the **additional experiments on real cryo-EM data (see Appendix E, figure 6, of the updated manuscript)** with very low signal-to-noise ratio suggested by the reviewer are convincing, we kindly ask you to consider raising the score.

---

### Official Review · Reviewer_4fc2 · 2024-11-04

**Soundness:** 2
**Presentation:** 2
**Contribution:** 2
**Rating:** 5
**Confidence:** 3

**Summary:**

The paper presents a theoretical framework aimed at addressing the limitations of existing self-supervised learning (SSL) methods for image reconstruction. However, several critical issues undermine its contributions and overall validity.
The proposed estimator builds on Stein’s Unbiased Risk Estimator (SURE) without offering significant advancements beyond existing methodologies. While the authors claim their approach allows for learning in scenarios with unknown noise structures, similar claims have been made in prior works without the need for extensive computational resources. This redundancy raises questions about the paper's originality and its contribution to the field.
The theoretical framework lacks clarity, making it difficult for readers to grasp the practical implications of the proposed methods. The authors introduce constrained self-supervised losses with Lagrange multipliers, but fail to provide sufficient detail on how these constraints improve learning outcomes compared to traditional methods. This omission hampers reproducibility and understanding, which are essential in scientific discourse.
Although the authors present evaluations claiming near state-of-the-art results, they do not adequately benchmark against a comprehensive set of competitive methods. Their focus on a narrow range of tasks, particularly in MRI reconstruction, limits the generalizability of their findings. The evidence provided does not convincingly demonstrate that their method consistently outperforms other self-supervised techniques across diverse applications.
The paper acknowledges that the proposed method requires more computational resources than existing approaches. This raises practical concerns regarding its applicability in real-world scenarios, especially considering that many researchers and practitioners may not have access to such resources. The trade-off between computational efficiency and performance is inadequately addressed, leaving potential users with unresolved questions about feasibility.
While the authors assert that their method generalizes well to complex tasks like MRI reconstruction, they provide insufficient evidence to substantiate these claims. Generalization is a critical aspect of any learning framework; without robust validation across varied datasets and tasks, such assertions remain speculative at best.
In summary, while the paper attempts to tackle relevant issues in self-supervised learning for image reconstruction, it ultimately falls short due to a lack of novelty, clarity, empirical validation, and robust generalization evidence. These shortcomings warrant rejection as they detract from the potential impact and applicability of the research within the broader field of machine learning and computer vision.

**Strengths:**

1)	The paper presents, a novel family of self-supervised estimators that effectively addresses the challenges of unknown noise levels, extending the capabilities of existing methods like SURE, and well-written.
2)	It offers a clear theoretical framework that elucidates the robustness-expressivity trade-off in self-supervised learning, enhancing the understanding of estimator dynamics.
3)	The paper includes thorough experiments on established datasets, providing solid empirical evidence for the effectiveness of their method compared to other self-supervised methods and showing its applicability across diverse imaging problems.

**Weaknesses:**

1) Computational complexity was already mentioned by the authors. Proposed method requires additional evaluations during training, leading to increased computational costs and potential implementation challenges.
2) The proposed ZED estimator struggles with high-entropy noise distributions, and the method's effectiveness may vary depending on optimizer settings and hyperparameter tuning. It would be better to provide possible strategies to remedy this
3) While the experiments demonstrate strong performance in specific tasks, there is limited exploration of diverse imaging applications and noise scenarios, which may affect generalizability.
4) The related work section can be summarized to better discuss the actual contribution of the current work.

**Questions:**

1)	In case of correlated Gaussian noise, the construction in (14) assumes that the set of s-dimensional covariance matrices includes the true covariance, i.e., the plausible covariance matrices. This intrinsically assumes a low dimensional structure, but this was not mentioned in the text. Also in practice, this assumption might not hold, and as such it would only do a low-dimensional projection of the true covariance onto s-dimensional space? The consequences and possible solutions of this case is not discussed.
2)	Table 1 results claim that the proposed method perform close (≈ 1dB) to the SURE with known noise model. However, 1dB in PSNR is a distinct difference, and claiming that performing close to the SURE is misleading.

---

> ### Author Response · Authors · 2024-11-19
> **Response to Reviewer 4fc2 (Part I)**
>
> Thanks for your comments. We believe that there might have been a misunderstanding regarding the two main concerns raised by the reviewer: i) additional computational complexity of the method and ii) applicability of the method to diverse imaging settings and high-entropy noise distributions. Please find below a detailed response that addresses these points.
>
> > While the authors claim their approach allows for learning in scenarios with unknown noise structures, similar claims have been made in prior works without the need for extensive computational resources. (...) The paper acknowledges that the proposed method requires more computational resources than existing approaches. This raises practical concerns regarding its applicability in real-world scenarios, especially considering that many researchers and practitioners may not have access to such resources. (...) Computational complexity was already mentioned by the authors. The proposed method requires additional evaluations during training, leading to increased computational costs and potential implementation challenges
>
>
> The proposed method requires **a single additional network evaluation for each training batch at most**, which is a relatively mild requirement in most practical cases. Note that other popular methods in the literature such as SURE or Neighbor2Neighbor also require two network evaluations per batch. Moreover, the proposed method requires a single network evaluation at test time, whereas other competing methods (e.g., Recorrupted2Recorrupted) require multiple evaluations. See a summary below:
>
> |          Method           | Evaluations Train | Evaluations Test | Any network Architecture |
> |:-------------------------:|:-----------------:|:---------------:|:------------------------:|
> |        Supervised         |         1         |        1        |           Yes            |
> |      UNSURE  (ours)       |         2         |        1        |           Yes            |
> |  UNSURE via score (ours)  |         1         |        1        |           Yes            |
> |           SURE            |         2         |        1        |           Yes            |
> |     R2R Noisier2Noise     |         1         |       >5        |           Yes            |
> |        Blind Spot         |         1         |        1        |            No            |
> | Noise2Void/ Noise2Inverse |        >1         |       >1        |           Yes            |
> |     Neighbor2Neighbor     |         2         |        1        |           Yes            |
>
> Thus, we disagree that "extensive computational resources" are needed, as the same computing ressources can be used to train existing methods (Recorrupted2Recorrupted, Noise2Self, etc.) and our proposed method.
>
> > While the authors claim their approach allows for learning in scenarios with unknown noise structures, similar claims have been made in prior works
>
>
> As shown in Figure 3 (right-hand side subfigures), the proposed method obtains better performance than the state-of-the-art method that handles unknown noise levels Noise2Score [*Noise distribution adaptive self-supervised image denoising using Tweedie distribution and score matching*, CVPR 2022].
>
> Moreover, to the best of our knowledge, no previous unsupervised method without access to the noise level (e.g., Noise2Self, Noise2Void, etc.) has been presented with theoretical guarantees. In this paper, we prove that the proposed method tracks the gold standard MMSE estimator.
>
> > The authors introduce constrained self-supervised losses with Lagrange multipliers but fail to provide sufficient detail on how these constraints improve learning outcomes compared to traditional methods
>
> The experiments in Section 5 of the paper demonstrate that the use of the proposed method with Lagrange multipliers obtains state-of-the-art performance. In particular, the results in Figure 3 show the practical gains of using Lagrange multipliers.
>
> Furthermore, the paper provides an exact theoretical characterization of the expected performance of the ZED denoiser in proposition 2. This explicitly relates it to the optimal MMSE estimator that theoretically can be learned in the supervised or SURE with known noise level settings.

---

> > ### Author Response · Authors · 2024-11-19
> > **Response to Reviewer 4fc2 (Part II)**
> >
> > > (...) they do not adequately benchmark against a comprehensive set of competitive methods. Their focus on a narrow range of tasks, particularly in MRI reconstruction, limits the generalizability of their findings. (...) While the experiments demonstrate strong performance in specific tasks, there is limited exploration of diverse imaging applications and noise scenarios, which may affect generalizability
> >
> > We propose a general loss that can be applied to any inverse problem, with theoretical guarantees that are independent of the dataset, which generalize across imaging modalities. The proposed method is **compared against the main self-supervised methods in the literature**, namely:
> >
> > - Noise2Score
> > - Noise2Void
> > - Noise2Inverse
> > - Neighbor2Neighbor
> > - SURE
> > - Recorrupted2Recorrupted
> >
> > The theoretical framework provided in the paper shows that the proposed method is expected to perform better than the methods above when the noise distribution is partially known, since it is more flexible than SURE and Recorrupted2Recorrupted to a mispecified noise level, while exploiting more information about the noise distribution than Noise2Score, Noise2Void, Noise2Inverse and Neighbor2Neighbor.
> >
> > Moreover, the experimental section provides **results for 5 different imaging scenarios**:
> > - Isotropic Gaussian denoising
> > - Spatially correlated Gaussian denoising
> > - Accelerated MRI reconstruction with Gaussian noise
> > - Sparse view computed tomography with Poisson-Gaussian noise.
> > - Real Cryo-EM data denoising (Appendix E of updated manuscript)
> >
> > Thus, there is no reason why the method would not perform well in other inverse problems.
> >
> > > The proposed ZED estimator struggles with high-entropy noise distributions, and the method's effectiveness may vary depending on optimizer settings and hyperparameter tuning. It would be better to provide possible strategies to remedy this
> >
> > There is a misunderstanding regarding this point. Section 3 states that the ZED estimator **fails with high-entropy signal distributions**, such as $p_x = \mathcal{N}(0, I\sigma_x^2)$, and not high-entropy noise distributions. Note that *all* unsupervised methods are expected to obtain poor denoising performances in this case, as it is impossible to distinguish the noise from the signal. For example, if both the signal and noise distributions are isotropic Gaussian, there is no way to distinguish noise from signal when the variance of the noise is unknown.
> >
> >
> > > In case of correlated Gaussian noise, the construction in (14) assumes that the set of $s$-dimensional covariance matrices includes the true covariance, i.e., the plausible covariance matrices. This intrinsically assumes a low dimensional structure, but this was not mentioned in the text.  Also in practice, this assumption might not hold, and as such it would only do a low-dimensional projection of the true covariance onto s-dimensional space? The consequences and possible solutions of this case is not discussed.
> >
> >
> > We believe there might be some confusion. For small $s$, the *family* of noise distributions is low dimensional in the same way that SURE assumes a single noise distribution (zero-dimensional family, $s=0$) or that an iid Gaussian model with unknown variance is a one-dimensional family, $s=1$. Note, however, that the *noise distribution itself is not low dimensional*, and it generally has high-entropy.
> >
> > The set of covariance matrices $\mathcal{R}$ considered in (14) is only a low dimensional family for the case of small $s$. The theory is equally applicable to arbitrary sets, up to considering any matrix covariance matrix as plausible, i.e., $s\approx n^2$. The paper shows that there is an inherent trade-off between the amount of prior knowledge about the covariance matrix and the performance of the estimator.
> >
> > The question of what happens when the true covariance does not belong to the assumed $s$-dimensional setting is evaluated empirically in Section 5 of the paper (Table 2). The experiment demonstrates that not having the true covariance in the set has a more severe effect on the performance than choosing a sufficiently large $s$ assuring that the covariance is included in the set.
> >
> > > Table 1 results claim that the proposed method performs close ($\approx$ 1dB) to the SURE with known noise model. However, 1 dB in PSNR is a distinct difference, and claiming that performing close to the SURE is misleading.
> >
> > We have modified the manuscript, removing the word "close" to avoid any confusion.

---

> > > ### Comment · Reviewer_4fc2 · 2024-12-01
> > >
> > > Sorry for my late response due to unforeseen circumstances.
> > >
> > > Thank you for the detailed responses in the rebuttal. After reviewing the comments of the other reviewers and the responses together with the paper, I will summarize my thoughts again as follows:
> > >
> > > The authors clarify that their method requires only one additional network evaluation per training batch, which is manageable compared to other methods that often require two or more. They argue that the computational demands are not excessive and can be handled with existing resources. They assert that their method outperforms state-of-the-art techniques like Noise2Score.
> > >
> > > They emphasize that their proposed loss function applies to a variety of inverse problems, supported by experiments across five imaging scenarios, including isotropic Gaussian denoising and accelerated MRI reconstruction. The authors address concerns about the ZED estimator's performance, clarifying that it struggles with high-entropy signal distributions rather than noise, as distinguishing between noise and signal becomes challenging under certain conditions.
> > >
> > > They discuss potential limitations regarding low-dimensional assumptions for covariance matrices but maintain that their theoretical framework applies broadly. Empirical evidence is provided on the impact of covariance assumptions on performance. Finally, the manuscript has been revised to remove potentially misleading language.
> > >
> > > While I acknowledge the improvements made by the authors, I still have reservations regarding the contribution and clarity with other points highlighted by other reviewers. Therefore, I will maintain my initial score.

---

> ### Author Response · Authors · 2024-12-02
>
> Many thanks for the feedback. We are happy to hear that the response **addressed the main concerns of the reviewer** regarding i) the computational complexity of the method, ii) the applicability of the method to a wide range of inverse problems, and iii) the broad scope of the theoretical framework presented in the paper, which covers general covariance matrices, not limited to low-dimensional families.
>
>  > While I acknowledge the improvements made by the authors, I still have reservations regarding the contribution and clarity with other points highlighted by other reviewers. Therefore, I will maintain my initial score.
>
> Could you please clarify what are the remaining concerns regarding the contribution and clarity? Which specific points raised by the other reviewers? Thank you!

---

### Meta-Review · Area_Chair_QrfG · 2024-12-27

**Metareview:**

The paper studies denoising with unknown noise level and/or distribution. It introduces an estimation framework inspired by SURE in which we minimize the mean squared self-prediction error subject to a zero-divergence constraint on the predictor. This can be converted into a min-max formulation, involving a divergence penalty. The paper studies extensions of this method to varying noise levels and correlated noise. The paper gives an expression for the expected MSE of the zero-divergence denoiser which minimizes the population MSE, in terms of the noise level and the (unconstrained) minimum MSE. When the latter is significantly smaller than \sigma^2 (as when the clean data x are structured), the ZED denoiser achieves near MMSE performance.

The main strengths of the paper are its novel formulation, which interpolates between SURE (in which the noise distribution is known) and methods such as noise2void and noise2noise which exhibit worse performance, but do not require detailed knowledge of the noise distribution. This situation is of practical importance, since in many applications, the noise can be modeled as independent across features, but with a-priori unknown variances. The proposed method also extends to signal-dependent noise such as Poisson. The paper demonstrates the advantages of this approach for both natural image denoising and medical image reconstruction. While the theoretical analysis is limited (as pointed out by the reviewer, it does not constitute an analysis of the UNSURE estimator per se, as it only pertains to the minimizer of the population MSE), it does provide preliminary corroboration of the method on structured signal distributions.

**Additional Comments On Reviewer Discussion:**

Reviewers generally found the paper to be well written, and observed that it improves over other denoising settings such as noise2noise and noise2void. The proposed method carries some of the benefits of SURE to the setting in which the noise standard deviation is unknowns, and potentially varying across features.

Reviewers raised a number of issues, including
- Computational complexity and high entropy noise distributions [4fc2]
- Certain limitations of the experiments [W3qG,4fc2]
- Technical concerns about proofs [K2y6]
- The strength of the theory [K2y6]

Issues around computational complexity and high entropy noise distributions were well addressed in the authors’ response. The method requires only one additional network evaluation per minibatch, and performs well when the *signal* distribution is low-entropy (i.e., when the unconstrained minimum mean squared error is significantly cannily smaller than \sigma^2).

Issues around the correctness of proofs [K2y6] were addressed in discussion.

The major point of criticism concerns the strength of the paper’s theoretical results. The paper’s analysis pertains only to the minimizer of the population version of the criterion, and does not address finite sample effects. As reviewer K2y6 points out, as theory, this falls short of the usual standards for analyzing statistical estimators (which pertain to finite sample size N and may focus on rates of convergence in N). If the sole contribution of this paper were theoretical, this limitation would put it below the bar. Weighing all of the contributions of the paper, including introducing a framework for denoising which generalizes SURE denoising to situations of unknown noise level / covariance, and validating it with theory (albeit preliminary) and experiments.

---

### Decision · Program_Chairs · 2025-01-22

Accept (Poster)